# Doubling multiplexed imaging capability via spatial expression pattern-guided protein pairing and computational unmixing
Gyuri Kim [1], Hyejin Shin[2], Minho Eom [1], Hyunwoo Kim [2], Jae-Byum Chang [2] ✉ &
Young-Gyu Yoon [1,3,4] ✉

Three-dimensional multiplexed fluorescence imaging is an indispensable technique in neuroscience. For two-dimensional multiplexed imaging, cyclic immunofluorescence, which involves repeating staining, imaging, and signal removal over multiple cycles, has been widely used. However, the application of cyclic immunofluorescence to three dimensions poses challenges, as a single staining process can take more than 12 hours for thick specimens, and repeating this process for multiple cycles can be prohibitively long. Here, we propose SEPARATE (Spatial Expression PAttern-guided paiRing And unmixing of proTEins), a method that reduces the number of cycles by half by imaging two proteins using a single fluorophore. This is achieved by labeling two proteins with the same fluorophores and unmixing their signals based on their three-dimensional spatial expression patterns, using a neural network. We employ a feature extraction network to quantify the spatial distinction between proteins, with these quantified values, termed feature-based distances, used to identify protein pairs. We then validate the feature extraction network with ten proteins, showing a high correlation between spatial pattern distinction and signal unmixing performance. We finally demonstrate the volumetric multiplexed imaging of six proteins using three fluorophores, pairing them based on feature-based distances and unmixing their signals through protein separation networks.

The development of bioimaging techniques in neuroscience has traditionally pursued two distinct strategies: three-dimensional (3D) imaging or multiplexed two-dimensional (2D) imaging. However, there is now a growing need to integrate both approaches into 3D multiplexed imaging. Considering that the brain is composed of more than a hundred different cell types[1], each precisely assembled within specific brain regions[2], it is clear that 2D multiplexed imaging can only capture a subset of cell types or their connectivity, often in an incomplete or incorrect context[3,4]. Therefore, 3D multiplexed imaging is essential to gain a deeper understanding of the brain. This need for multiplexed 3D imaging is not limited to neuroscience; recently, various studies have reported that 2D multiplexed imaging can provide only a subset of the total information available and can sometimes be misleading[5–7]. Multiplexed 3D imaging is vital in cancer biology to better understand cancer heterogeneity and improve diagnosis[8]. However, despite significant efforts, multiplexed 3D imaging remains challenging, as it must address multiple complex problems[9].

The most common means of achieving multiplexed protein imaging is via cyclic immunofluorescence, which repeats the staining of targets with antibodies, image acquisition, and antibody removal or fluorophore inactivation[10]. In cyclic immunofluorescence, the number of fluorophores that can be used simultaneously is typically limited to two or three due to spectral overlap[11,12]. As a result, to image ten different proteins, a total of four cycles of staining, imaging, and signal removal are required. In 2D imaging of thin tissue slices, such as those with a thickness of less than 10 μm, where staining can be completed relatively quickly—even in about an hour—the repeated staining process is not a major issue. However, specimens with a thickness greater than 10 μm, such as 35 μm, typically require substantially longer staining times, often extending up to 10 h with conventional

[1]School of Electrical Engineering, KAIST, Daejeon, Republic of Korea. [2]Department of Materials Science and Engineering, KAIST, Daejeon, Republic of Korea. [3]Department of Semiconductor System Engineering, KAIST, Daejeon, Republic of Korea. [4]KAIST Institute for Health Science and Technology, Daejeon, Republic of Korea. ✉e-mail: jbchang03@kaist.ac.kr; ygyoon@kaist.ac.kr

immunostaining approaches that use full IgG antibodies[13]. As a result, the total staining time using conventional methods for ten-color imaging could exceed several days. Prolonged staining time does not merely extend the total process; it can also compromise specimen quality and antigen integrity[14,15]. Spectral imaging and unmixing could provide a solution to this problem, as they allow the use of more than three fluorophores in a single staining cycle[16]. However, spectral imaging can be complex and often requires specialized equipment capable of acquiring images from various spectral ranges[17,18]. Therefore, there is an urgent need for techniques that can reduce the total staining time for 3D multiplexed imaging.

To overcome these challenges, we propose SEPARATE (Spatial Expression PAttern-guided paiRing And unmixing of proTEins), a volumetric multiplexed imaging technique that allows us to capture signals from two proteins simultaneously in a single image that are unmixed through machine learning. To ensure the accurate unmixing of fluorescent signals from distinct proteins, we first identify pairs of proteins with distinct spatial expression patterns through contrastive learning using a convolutional neural network, referred to as the feature extraction network. We then employ another convolutional neural network, the protein separation network, which takes the image of two proteins labeled with one fluorophore as input and predicts two images, each containing only one protein. We demonstrate to what degree the quantified distinction between proteins derived from the feature extraction network (feature-based distance) matches the performance of the protein separation network. Finally, we demonstrate the volumetric multiplexed imaging of six proteins using three fluorophores in a mouse brain in a single staining and imaging session.

## Results

### SEPARATE: pairing and unmixing proteins with distinctive spatial expression patterns

SEPARATE enables volumetric multiplexed imaging by addressing the challenge of algorithmic unmixing, that certain proteins exhibit visually similar spatial expression patterns. The first step is to identify pairs of proteins with distinct spatial expression patterns. Each pair is then labeled and imaged using a single fluorophore, and subsequently, the images are unmixed into two individual images, each containing information from one protein, through machine learning. This framework allows us to capture the 3D signals of 2N proteins using only N fluorophores in a single imaging round, in contrast to conventional methods that require one fluorophore per protein per imaging round.

Figure 1a illustrates a representative example of how SEPARATE works with four proteins, referred to as A, B, C, and D. These four proteins can be paired in two different ways to image two proteins using the same fluorophore. First, A and B are paired, and C and D are paired. Second, A and C are paired, and B and D are paired. In the first option, one pair consists of two fibrous proteins and the other pair consists of two globular proteins. In the second option, each pair consists of one fibrous protein and one globular protein, in which each protein is more visually distinguishable. Thus, it is reasonable to predict that the protein separation network will perform better in the second way.

However, manual identification of the optimal grouping is infeasible due to the diverse spatial expression patterns across different locations and proteins. We address this by employing a feature extraction network that quantifies visual distinctiveness and finds the optimal grouping in an automated fashion. The network is trained through contrastive learning to maximize an interclass distance and minimize an intraclass distance[19] (Fig. 1b). The interclass distance $d_{inter}$ represents the average distance between the feature vectors of different proteins, and the intraclass distance $d_{intra}$ represents the average distance between the feature vectors within the same proteins, which can be mathematically expressed as follows:

$$d_{inter} = \frac{1}{|\mathbf{P}|(|\mathbf{P}|-1)} \sum_{\substack{p_1 \in \mathbf{P} \\ p_1 \neq p_2}} \sum_{p_2 \in \mathbf{P}} ||\bar{f}_{p_1} - \bar{f}_{p_2}||$$

$$d_{intra} = \frac{1}{|\mathbf{P}|} \sum_{p \in \mathbf{P}} \frac{1}{N_p} \sum_{i=1}^{N_p} ||\bar{f}_p - f_p^{(i)}||,$$

where $\mathbf{P}$ is a set of proteins, $|\mathbf{P}|$ is the number of proteins in set $\mathbf{P}$, $f_p^{(i)}$ is a feature vector of protein $p$, $\bar{f}_p$ is the mean of feature vectors $f_p^{(i)}$ (center of the feature vectors), $N_p$ is the number of feature vectors $f_p^{(i)}$, $||\cdot||$ is the average of L1 and L2 norms.

This contrastive learning approach ensures that a distinct cluster of feature vectors is formed for each protein (Fig. 1c). Clusters are constructed in relative locations based on their spatial expression patterns, with similar proteins placed close together and distinct proteins positioned further apart. As a measure of the distinction of spatial expression patterns, distances between every pair of two protein clusters, termed feature-based distances, are calculated (Fig. 1d).

These feature-based distances are then utilized as a precursor for predicting unmixing performance between protein pairs, guiding identify optimal grouping of protein pairs that can be effectively distinguished (Fig. 1e). Specifically, we search for the optimal grouping $G_{opt}$ defined as follows:

$$G_{opt} = \underset{G}{\mathrm{argmax}} \min(d_1, d_2, \ldots, d_n),$$

where G represents the possible grouping of $2n$ proteins into $n$ pairs, and the $d_i$ denotes the feature-based distance between the two proteins in the $i$-th pair ($i = 1, 2, \ldots n$). Our criterion for identifying optimal grouping by maximizing the minimum distance among paired proteins ensures robust unmixing, even for the most challenging protein combinations. This approach emphasizes consistent performance across all protein pairs rather than allowing exceptional unmixing in some pairs at the expense of others.

Once the optimal grouping is identified, we experimentally obtain the three-channel images of each pair for training the protein separation network. We use three different fluorophores to label two proteins, α and β: one for protein α, another for protein β, and the third for both proteins α and β (Fig. 1f). Consequently, each channel of the image contains the fluorescent signal of protein α, the fluorescent signal of protein β, and the mixed fluorescent signal of proteins α and β, respectively (Fig. 1g).

We note that the relative signal intensity of proteins α and β in the mixed image can vary significantly due to multiple experimental factors, including the concentration of each antibody and the level of protein expression. This variability can undermine the robustness of the protein separation network trained on real data, where the mixed image is used as the input and the other images as targets. To address this issue, instead of attempting to match the relative brightness in the training and test data by optimizing the antibody concentration through repetitive imaging experiments, we generated synthetic images. These synthetic images are created by linearly superimposing the images of each protein with a random ratio and are used to construct the training dataset, thereby making the protein separation network robust to variations in relative signal intensity (Fig. 1h, i).

### Validation of the feature extraction network's predictive power for unmixing performance

In order to verify the effectiveness of the distance measured using the feature extraction network in predicting the unmixing performance of protein separation networks, we first acquired 3D immunofluorescence images of the dentate gyrus (DG) region in mouse brain slices, each stained with antibodies against one of ten proteins, such as calbindin 2, calnexin, doublecortin, glial fibrillary acidic protein (GFAP), lamin B1, microtubule-associated protein 2 (MAP2), neuronal nuclei (NeuN), nucleolin, parvalbumin (PV), and S100B (Fig. 2a). Thereafter, the feature extraction network was trained using these images. Figure 2b shows the t-SNE plot of the feature vectors extracted by the trained network and single dot in the t-SNE plot represents a feature vector extracted from a single image patch of protein.

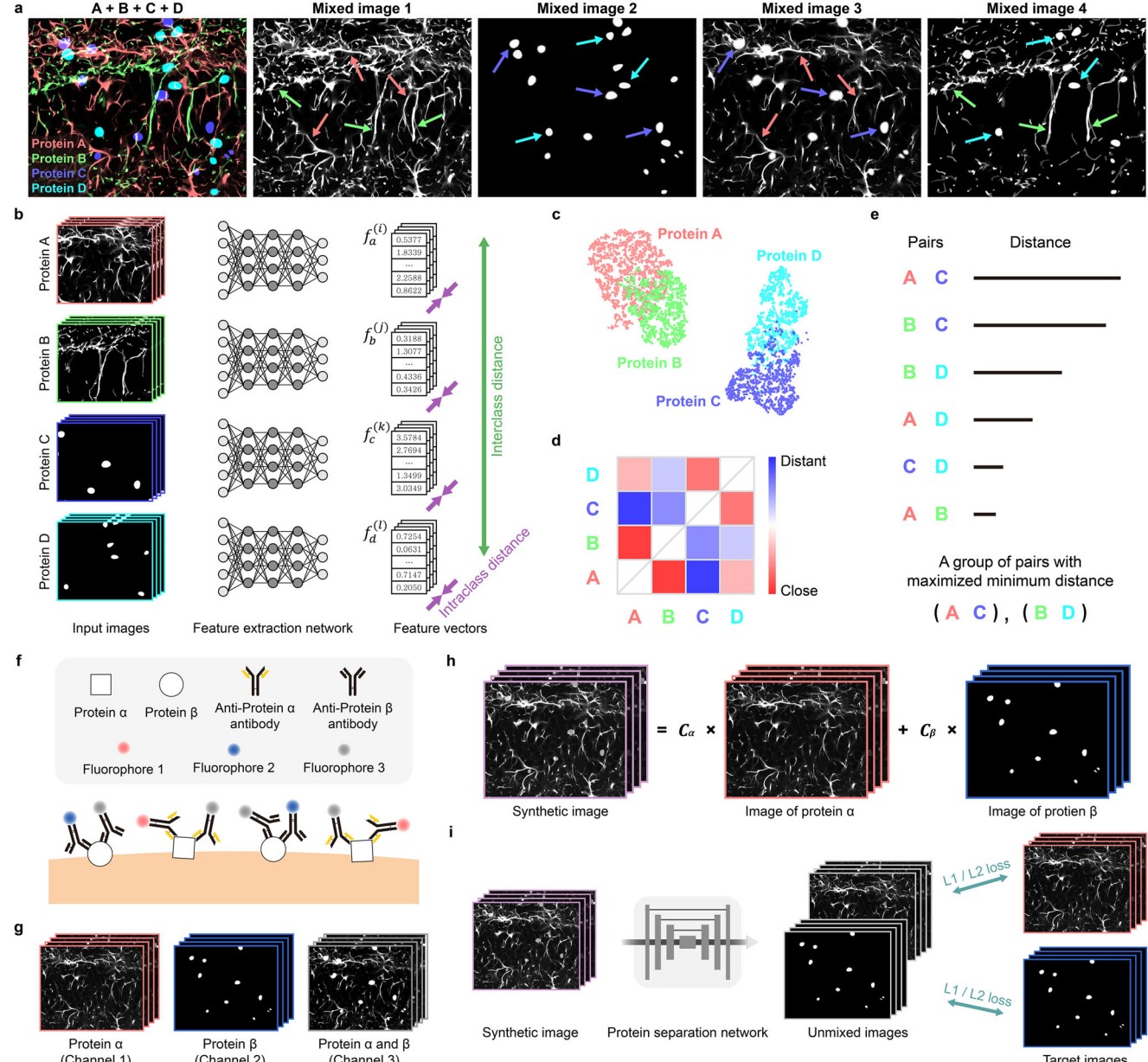

**Fig. 1 | The overall framework of SEPARATE. a** An illustrative example of imaging four proteins (A, B, C, D) using two fluorophores. Proteins are grouped as (A, B) and (C, D) in mixed image 1 and 2, and as (A, C) and (B, D) in mixed image 3 and 4, respectively. Proteins are more distinguishable in the second grouping. **b** Training scheme of the feature extraction network. The feature extraction network is trained to extract the feature vector $f_p^{(i)}$ of protein $p$ through contrastive learning by maximizing the interclass distance and minimizing the intraclass distance. **c** Formation of distinct clusters of extracted feature vectors for each protein using the feature extraction network, with clusters based on spatial expression pattern. Similar proteins are placed close together, and distinct proteins are positioned further apart. **d** A matrix showing the distance between clusters of feature vectors for each pair of proteins. These feature-based distances are used to identify the optimal grouping of protein pairs. **e** Identification of the optimal grouping of protein pairs by maximizing the minimum distance of protein pairs within the group. **f** Immunostaining using three fluorophores to label two proteins for training protein separation network. Fluorophore 1 and 2 are tagged to protein α and protein β, respectively. Fluorophore 3 is tagged to both proteins α and β. For simplicity, secondary antibodies are not shown in this scheme. **g** Obtained three-channel immunofluorescence image stack. Channel 1 displays protein α corresponding to the signal of fluorophore 1, and Channel 2 displays protein β corresponding to the signal of fluorophore 2. Channel 3 displays both proteins α and β, corresponding to the signal of fluorophore 3. **h** Generation of the synthetic image for training protein separation network. The synthetic image is generated by linearly combining images of protein α (Channel 1) and protein β (Channel 2) with the coefficient $C_\alpha$ and $C_\beta$, instead of optimizing the antibody concentration through repetitive imaging experiment. **i** Training scheme of the protein separation network. The protein separation network is trained to predict unmixed images of two proteins from synthetic image, by minimizing both the L1 loss and L2 loss between the unmixed images and target images.

Proteins exhibiting similar spatial expression patterns were located closely in the t-SNE plot. For example, a certain protein pair, nucleolin and NeuN, clusters together due to their similar spatial expression patterns, which are highly localized to the nuclei. Likewise, two proteins that are expressed in a fibrous pattern, GFAP and doublecortin, were closely positioned in the t-SNE plot. As shown in Supplementary Figs. 1 and 2, image patches corresponding to the overlapping region of the t-SNE plot are likely indistinguishable, whereas patches from the non-overlapping regions show more distinguishable patterns. In the case of nucleolin and NeuN, image patches from the overlapping region of the two clusters show lower cell density, where the information available to distinguish two proteins is limited. Conversely, moving further away from the overlapping region, image

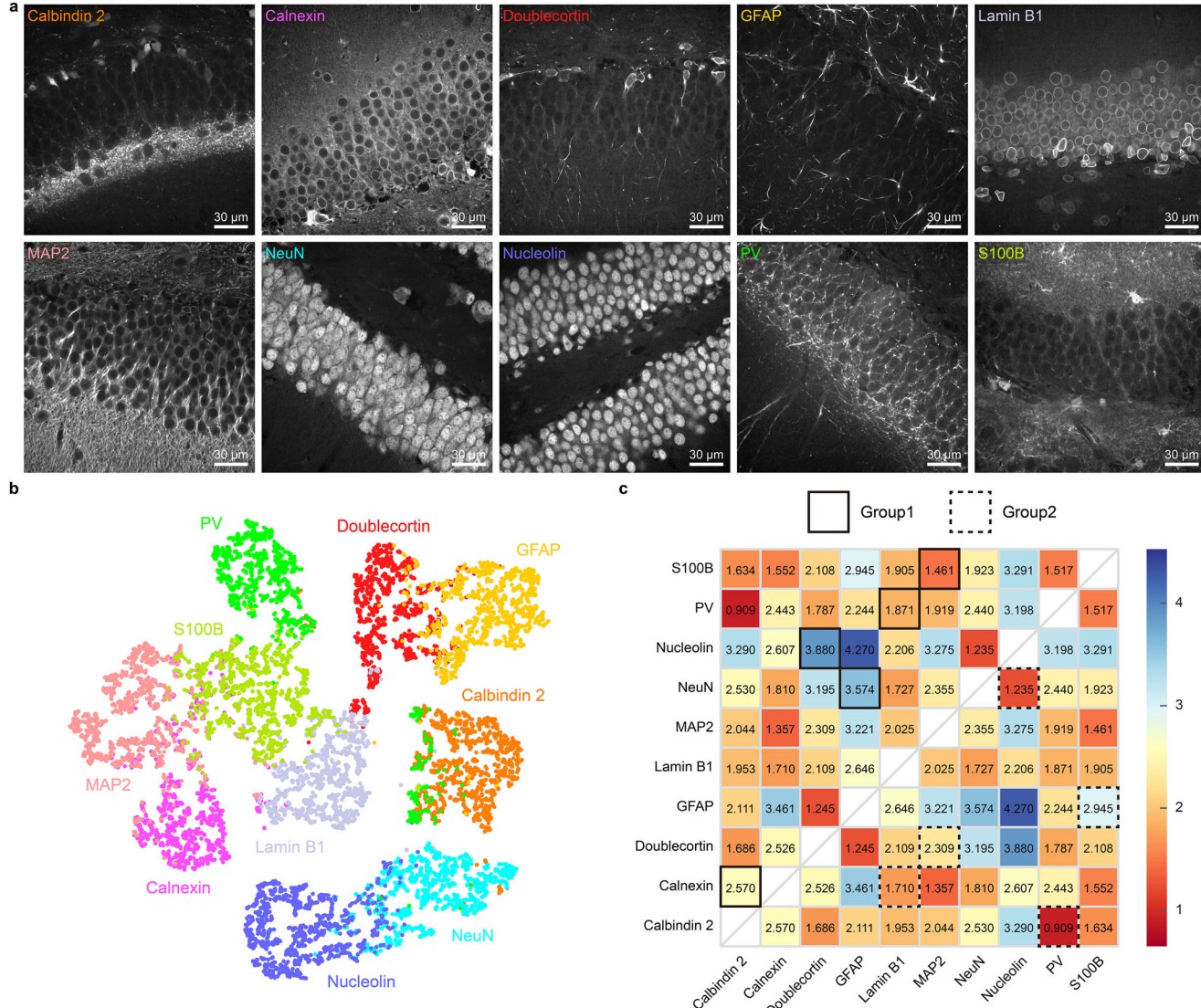

**Fig. 2 | Feature vector extraction and feature-based distance. a** Representative images of ten proteins: calbindin 2, calnexin, doublecortin, GFAP, lamin B1, MAP2, NeuN, nucleolin, PV, and S100B. **b** Two-dimensional visualization of feature vectors using t-SNE. Each dot represents a feature vector extracted from a single image patch of protein. The feature vectors form clusters for each protein, and proteins with similar spatial expression patterns are located near each other. **c** Pairwise feature-based distance matrix is displayed as heatmap with a red–yellow–blue color scheme, where blue indicates higher and red indicates lower feature-based distances. The feature-based distances are computed between clusters of feature vectors for each protein pair. Ten proteins are grouped in two ways, with pairs from each group highlighted on the matrix. The optimal grouping, group 1, achieves a minimum distance of 1.4609, while the alternative grouping, group 2, yields a minimum distance of 0.9092. Scale bar = 30 μm in (**a**).

patches exhibit higher cell density, making it easier to clearly observe the difference in the spatial expression pattern of two proteins. As a measure of the distinction between the spatial expression patterns of proteins, we used the average linkage distance between each pair of clusters, calculated as an average of all pairwise Euclidean distances between the point in one cluster and the point in the other cluster. The distance matrix between every pair of clusters of proteins is presented in Fig. 2c.

We grouped these ten proteins in two different ways, each group consisting of five pairs. The optimal grouping, group 1 consists of the following five pairs: (calbindin 2, calnexin), (doublecortin, nucleolin), (GFAP, NeuN), (lamin B1, PV), and (MAP2, S100B). The alternative grouping, group 2 consists of the following pairs: (calbindin 2, PV), (calnexin, lamin B1), (doublecortin, MAP2), (GFAP, S100B), and (NeuN, nucleolin). We then stained mouse brain slices and obtained 3D three-channel images for each pair of two proteins, including two individual signals of each protein and one mixed signal of both proteins. For example, a mouse brain slice was stained with a mixture of antibodies to show the calbindin 2 signal in the 488 channel, the calnexin signal in the 561 channel, and both calbindin and calnexin in the 647 channel. We then fed the mixed image acquired in the 647 channel to the corresponding protein separation network to unmix the signals of the two proteins. Figure 3a, b present the unmixed results for each protein pair in group 1 and group 2, respectively, merged by channel dimension using a green-and-magenta color scheme. Supplementary Figs. 3–12 provide a more detailed view, showing the input images alongside the unmixing results and ground truth images, with each channel displayed separately as individual grayscale images for better comparison of signal distribution and intensity.

We assessed the unmixing performance by comparing the output images with the ground truth images using the structural similarity index measure[20] (SSIM) and Pearson correlation coefficient (PCC). We then compared the unmixing performance with the feature-based distance (Fig. 3c). The SSIM and PCC were calculated for 16 non-overlapping 3D patches, treating each patch as an independent sample, and represented as a single data point in a box-and-whisker plot. For each protein pair, the

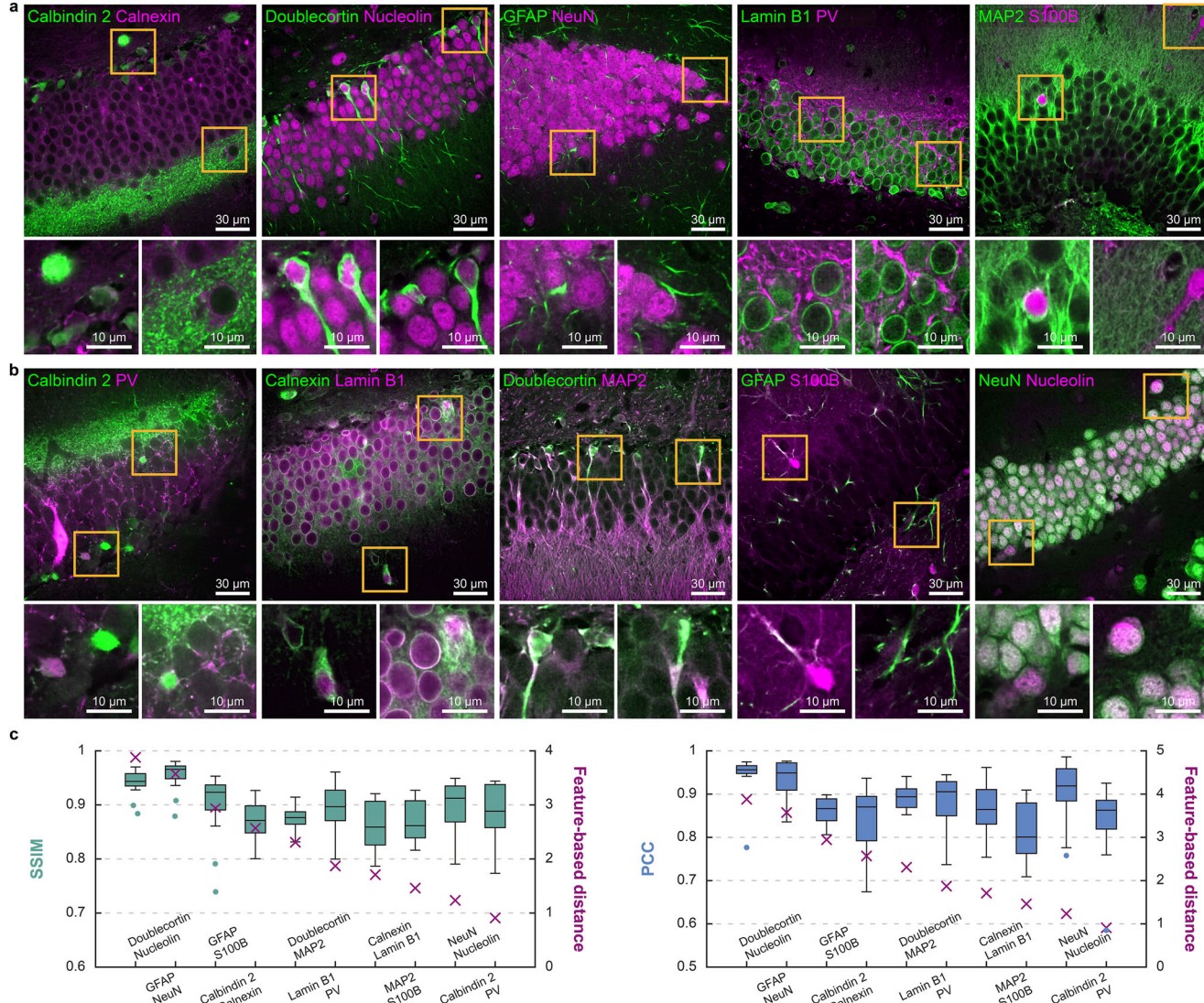

**Fig. 3 | Experimental validation of protein pairing using feature extraction network for fluorescent signal unmixing performance. a, b** Unmixing results of two groups shown in channel-wise merge using a green-and-magenta color scheme. Magnified views of regions highlighted by yellow boxes are shown below each pair; **a** Group 1, optimal grouping, consists of (calbindin 2, calnexin), (doublecortin, nucleolin), (GFAP, NeuN), (lamin B1, PV), and (MAP2, S100B); **b** Group 2, alternative grouping, consists of (calbindin 2, PV), (calnexin, lamin B1), (doublecortin, MAP2), (GFAP, S100B), and (NeuN, nucleolin). **c** Comparison of unmixing performance (SSIM and PCC) and feature-based distances. Unmixing performance was calculated between ground truth and unmixed images for non-overlapping patches ($n = 16$) and displayed as box-and-whisker plots with the feature-based distances shown as a purple x marker. The boxes show the interquartile range (IQR) with the median, while whiskers extend to 1.5 times the IQR. Individual points represent outliers, defined as values falling outside the whiskers. Protein pairs were rearranged in order of decreasing feature-based distance values, with the feature-based distance showing Pearson correlation coefficients of 0.7144 with average SSIM, 0.6873 with median SSIM, 0.6212 with average PCC, and 0.6024 with median PCC. Scale bar = 30 μm in full-field images and 10 μm in magnified views in (**a, b**).

unmixing performance was calculated separately for individual proteins and then averaged for representation in Fig. 3c. The unmixing performance for each individual protein is shown in Supplementary Fig. 13. Unmixing results of each patch with SSIM values are visualized in Supplementary Figs. 14 and 15. In line with our prediction, protein pairs with a larger distance between their clusters in the feature domain were more accurately unmixed, with the feature-based distance showing Pearson correlation coefficients of 0.7144 with average SSIM, 0.6873 with median SSIM, 0.6212 with average PCC, and 0.6024 with median PCC.

To account for inherent spatial similarities between proteins such as NeuN and nucleolin which both show nuclear localization, we introduced a normalized performance metric that adjusts the raw unmixing performance by the cross-similarity between protein pairs (Supplementary Fig. 16). This normalization provides a more accurate assessment of unmixing quality by controlling for cases where proteins naturally share similar spatial distributions. Notably, the correlation between feature-based distance and normalized unmixing performance was substantially stronger than with raw performance metrics: Pearson correlation coefficient of 0.9404 for average normalized SSIM and 0.7699 for average normalized PCC.

To statistically validate our approach, we compared the unmixing performance of proteins when placed in optimal versus alternative groupings (Supplementary Fig. 17). Using two-sided paired-sample *t* tests, we found that proteins in optimal groupings consistently exhibited higher unmixing performance where significant differences were observed. We further assessed normalized unmixing performance, which accounts for the inherent cross-similarity between paired proteins. This normalized analysis revealed more pronounced statistical differences between optimal and alternative groupings.

This result demonstrates that predicting protein pairs for optimal unmixing accuracy is feasible by selecting proteins positioned far apart in

the feature domain generated from singly stained specimens. For instance, when working with ten proteins, instead of experimentally testing all 45 possible combinations for unmixing accuracy, we can efficiently determine optimal pairs by simply training the feature extraction network with images of the 10 singly stained proteins, significantly reducing the complexity.

### Robustness of the feature-based distance across experimental variations

To validate the robustness of feature-based distance across various experimental conditions, we investigated how feature-based distances vary under different experimental settings. We analyzed the relationships between distance matrices rather than directly comparing optimal groupings. This approach was chosen because the relative relationships and trends between feature-based distances are more informative than their absolute values, as the feature extraction network incorporates randomness during the training process to ensure robustness.

We first compared distance matrices derived from separate training and test datasets, which revealed highly consistent patterns with the Pearson correlation coefficient of 0.9923, demonstrating stability across different samples (Supplementary Fig. 18). Additionally, through cross-validation analysis using four independent data, we evaluated the stability by rotating these data between training and testing roles (Supplementary Fig. 19). The Pearson correlation coefficients between distance matrices from different experiments consistently exceeded 0.85, confirming that our approach reliably captures intrinsic protein relationships regardless of dataset combinations.

To further assess the effect of imaging conditions, we evaluated spatial resolution by applying progressive binning factors (2, 4, 8, 16, 32, and 64) to our dataset (Supplementary Fig. 20). The matrices showed high correlation values exceeding 0.85 with the original resolution up to a binning factor of 16, indicating preservation of essential protein expression patterns. However, correlation values fell below 0.7 at binning factors of 32 and 64, where cellular structures became severely pixelated, defining a practical threshold for reliable feature extraction.

We also implemented a self-supervised denoising technique to ensure feature extraction reflects spatial patterns rather than experimental variations. Analysis across different noise conditions showed that denoised images maintained correlation values around 0.9 even with added Gaussian noise, while the raw noisy data showed correlations below 0.8 (Supplementary Fig. 21). This demonstrates that our preprocessing effectively normalizes noise characteristics across protein images.

These validations confirm that our method maintains consistency across different samples, imaging resolutions, and noise levels, providing a reliable foundation for protein expression pattern analysis.

### Demonstration of the volumetric multiplexed image of six proteins

Based on this result, we finally demonstrated volumetric multiplexed imaging of six proteins—doublecortin, GFAP, lamin B1, NeuN, nucleolin, and PV—using only three fluorophores with SEPARATE, achieving an imaged volume thickness of 25 μm in tissue sections. We trained the feature extraction network only for these six proteins and examined their feature-based distances (Fig. 4a). Based on this analysis, we identified the optimal grouping that maximizes the minimum distance among the three pairs as follows: (GFAP, NeuN), (lamin B1, PV), and (doublecortin, nucleolin). After selecting three pairs, we used three different fluorophores to obtain 3-color fluorescence images of six proteins. GFAP and NeuN were labeled with CF488A, lamin B1 and PV with CF568, and doublecortin and nucleolin with CF633. The images of each channel are displayed in Fig. 4b. We then unmixed the signal into individual signals of each protein using the protein separation network. The unmixing results of each channel are shown in Fig. 4c–e. In addition, the 3D rendering of the combined result is shown in Fig. 4f, demonstrating the capability of SEPARATE as a volumetric multiplexed imaging method.

### Extended validation demonstrates broad applicability and generalizability of SEPARATE

To further demonstrate the broad applicability of our method, we extensively validated SEPARATE using a public human cell dataset from Allen Institute for Cell Science[21]. The dataset contains 10 proteins: alpha-tubulin, desmoplakin, lamin b1, myosin IIB, Sec61 translocon beta subunit (Sec61β), ST6 beta-galactoside alpha-2,6-sialyltransferase 1 (ST6GAL1), translocase of outer mitochondrial membrane 20 (TOMM20), tight junction protein 1 (ZO-1), cell membrane labeled with CellMask, and DNA labeled with Hoechst (Fig. 5a). This dataset provides image stacks where cell membrane and DNA are imaged together with one of the other proteins in the same field of view.

Using the feature extraction network, we obtained feature vectors for each protein and visualized them in two dimensions using t-SNE (Fig. 5b). Based on the pairwise feature-based distances between protein clusters, we determined the optimal grouping that maximized the minimum distance between paired proteins: (alpha-tubulin, DNA), (desmoplakin, ZO-1), (lamin B1, cell membrane), (myosin IIB, ST6GAL1), and (Sec61β, TOMM20). The optimal pairs are highlighted by dashed boxes in the pairwise feature-based distance matrix (Fig. 5c). Notably, these optimal grouping successfully avoided combining proteins with similar spatial expression patterns. Membrane-associated proteins ZO-1 and cell membrane were assigned to different pairs, and cytoskeletal proteins alpha-tubulin and myosin IIB were also assigned to different pairs. Although desmoplakin and ZO-1 are both membrane-associated proteins, they show distinct spatial expression patterns as desmoplakin exhibits punctate patterns at desmosomes while ZO-1 shows fibrous patterns at tight junctions.

Since this dataset provides image stacks where cell membrane and DNA are each imaged together with one of the other proteins in the same field of view, we trained protein separation networks for all possible 16 pairs: 8 pairs of cell membrane with each of the other proteins and 8 pairs of DNA with each of the other proteins. Supplementary Figs. 22 and 23 show the unmixing results in channel-wise merge using a green-and-magenta color scheme of proteins paired with cell membrane and DNA, respectively. As this dataset does not contain real mixed images, we used synthetic mixed images for visualization of unmixing results and unmixing performance calculation. We used 0.5 for both coefficients, which represents an equal contribution from each protein image in synthetic mixed images.

We then analyzed the relationship between these feature-based distances and unmixing performance, quantified by SSIM between ground truth and unmixed images. The analysis was performed on two sets of protein pairs: eight pairs consisting of cell membrane with each of the other proteins (Fig. 5d) and eight pairs consisting of DNA with each of the other proteins (Fig. 5e). Unmixing performance was calculated with n = 11 for each pair and shown as box-and-whisker plots. This analysis showed that protein pairs with larger feature-based distance achieved more accurate unmixing, evidenced by the Pearson correlation between feature-based distances and unmixing performances. For proteins paired with cell membrane, feature-based distances show Pearson correlation coefficients of 0.7800 with average SSIM, 0.8096 with median SSIM, 0.6481 with average PCC, and 0.6317 with median PCC. For proteins paired with DNA, feature-based distances show Pearson correlation coefficients of 0.8825 with average SSIM, 0.8642 with median SSIM, 0.6078 with average PCC, and 0.5521 with median PCC. The unmixing performance for each individual protein is shown in Supplementary Fig. 24. These results demonstrate that our approach performed robustly on existing dataset that is acquired from different species (human) and imaging procedures, highlighting the method's generalizability.

### Discussion

In this study, we present SEPARATE, a method for volumetric multiplexed fluorescence imaging that unmixes signals of two proteins from a single fluorophore through deep learning by leveraging diverse spatial expression

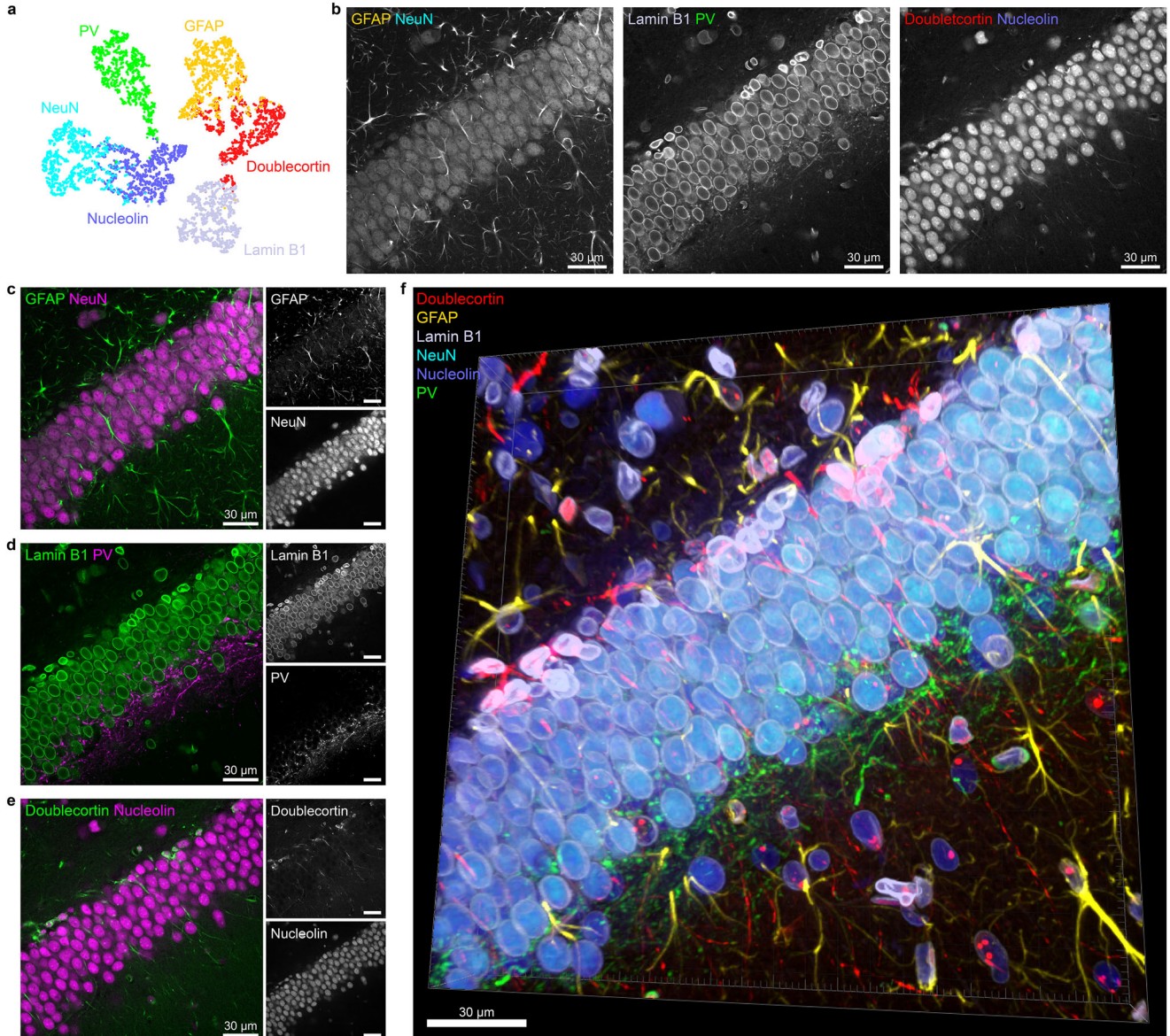

**Fig. 4 | Demonstration of volumetric multiplexed image of six proteins. a** The t-SNE plot of extracted feature vectors of six proteins—doublecortin, GFAP, lamin B1, NeuN, nucleolin, and PV— used for demonstration. **b** Three-channel stained images of six proteins. Each image represents the mixed signal of two proteins corresponding to each pair determined based on feature-based distances. **c–e** Channel-wise merged unmixed images and individual images for each channel. **c** GFAP and NeuN; **d** lamin B1 and PV; **e** doublecortin and nucleolin. **f** The three-dimensional rendering of the multiplexed image for six proteins. Scale bar = 30 μm in (**b–f**).

patterns, thus allowing for fewer fluorophores than the number of proteins. SEPARATE reduces the number of imaging cycles required for imaging a given number of proteins by up to half, thereby significantly decreasing overall process time and complexity.

A key insight from our research is the importance of selecting protein pairs with distinct spatial expression patterns for effective signal unmixing. To demonstrate that the ability to unmix the signal relies on the distinguishability of the spatial expression patterns of paired proteins, we employed the feature extraction network. Through the feature extraction network, we quantified the distinctiveness of spatial expression patterns inherent in each protein, enabling the identification of optimal grouping. Notably, the use of 3D images of stained specimens to train the protein separation network was crucial, as many proteins' spatial expression patterns are not clearly identifiable in 2D images (Supplementary Figs. 25 and 26). For instance, fibrous proteins such as GFAP or doublecortin appear as dots or short lines in 2D images, whereas their fibrous

structures are more clearly identifiable in 3D images. By utilizing 3D images for training the protein separation network, we achieved a high level of unmixing accuracy.

To validate our approach, we tested SEPARATE with ten proteins and found that the feature-based distance between proteins correlates with unmixing performance, with larger distances between clusters yielding more effective unmixing as evidenced by higher SSIM scores. This relationship validates the effectiveness of the feature extraction network as a precursor of unmixing success. We also validated the robustness of the feature-based distance across experimental variations (Supplementary Figs. 18–21), and constraints. Especially, we conducted Monte Carlo simulations to evaluate scalability with larger marker sets (Supplementary Fig. 27). Since our feature vectors are latent variables, whose absolute values are less meaningful than their relative relationships, we normalized the feature space to constrain all vectors within a unit circle for these simulations. In this normalized feature space, our experiments showed that protein

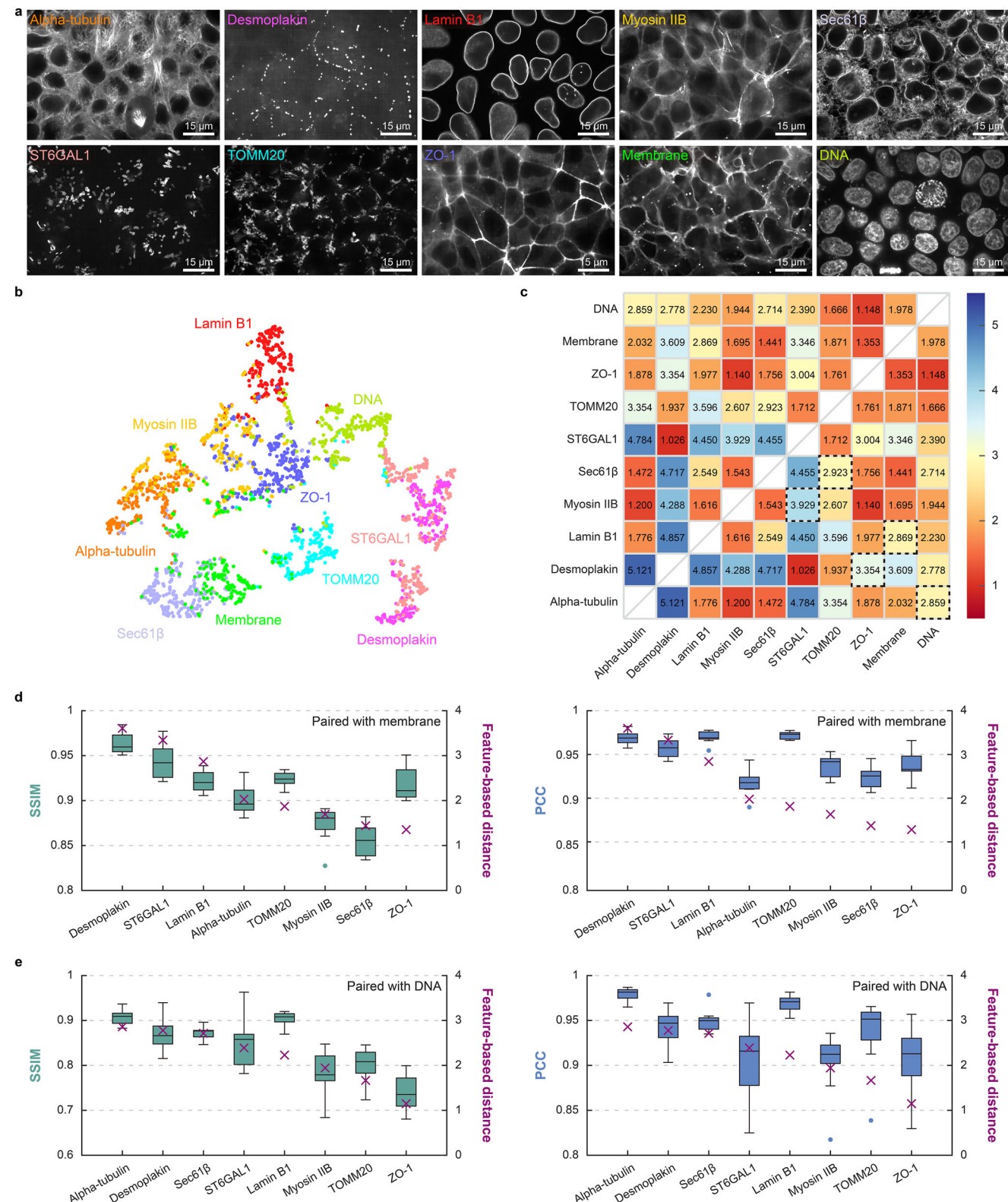

pairs with normalized feature-based distance greater than 1.5 consistently yielded unmixing performance with SSIM values above 0.9.

Building on this validation, we demonstrated the volumetric multiplexed imaging of six proteins using only three fluorophores. By automatically pairing proteins based on their feature-based distances, we achieved clear, unmixed signals, effectively doubling the number of proteins that could be simultaneously visualized with the available fluorophores.

Further validation using the public human cell dataset from Allen Institute for Cell Science corroborated these findings, demonstrating that feature-based distance serves as a reliable precursor of unmixing performance across diverse cellular contexts.

SEPARATE's practical implementation follows a two-phase approach. The initial setup phase involves several steps: immunofluorescence staining for protein pairing, feature extraction network training to identify optimal

**Fig. 5 | Extended validation of SEPARATE on the public human cell dataset from Allen Institute for Cell Science. a** Representative images of ten proteins in the public human cell dataset from Allen Institute for Cell Science: alpha-tubulin, desmoplakin, lamin B1, myosin IIB, Sec61β, ST6GAL1, TOMM20, and ZO-1. **b** Two-dimensional visualization of feature vectors using t-SNE. Each dot represents a feature vector extracted from a single image patch of protein. **c** The pairwise feature-based distance matrix computed between clusters of feature vectors for each protein pair and displayed as a heatmap with a red-yellow-blue color scheme, where blue indicates higher and red indicates lower feature-based distances between protein pairs. **d** Comparison of unmixing performance (SSIM and PCC) and feature-based distances for 8 proteins paired with cell membrane. Unmixing performance was calculated between ground truth and unmixed images for each image stack ($n = 11$ per pair) and displayed as box-and-whisker plots with the feature-based distances shown as a purple x marker. The boxes show the interquartile range (IQR) with the median, while whisker extends to 1.5 times the IQR. Individual points represent outliers, defined as values falling outside the whiskers. Protein pairs were rearranged in order of decreasing feature-based distance values. Protein pairs with larger feature-based distances showed more accurate unmixing, with feature-based distances showing Pearson correlation coefficients of 0.8800 with average SSIM, 0.8096 with median SSIM, 0.6481 with average PCC, and 0.6317 with median PCC across z-stacks. **e** Comparison of unmixing performance (SSIM and PCC) and feature-based distances for 8 proteins paired with DNA. Unmixing performance was calculated between ground truth and unmixed images for each image stack ($n = 11$ per pair) and displayed as box-and-whisker plots with the feature-based distances shown as a purple x marker. The boxes show the interquartile range (IQR) with the median, while whisker extends to 1.5 times the IQR. Individual points represent outliers, defined as values falling outside the whiskers. Protein pairs were rearranged in order of decreasing feature-based distance values. Protein pairs with larger feature-based distances showed more accurate unmixing, with feature-based distances showing Pearson correlation coefficients of 0.8825 with average SSIM, 0.8642 with median SSIM, 0.6078 with average PCC, and 0.5521 with median PCC across z-stacks. Scale bar = 15 μm in (**a**).

grouping, specialized training data preparation with separate protein channels, and protein separation network training. Both immuno-fluorescence staining processes require standard staining time, training the feature extraction network for identifying optimal grouping takes approximately 1 h, and training the protein separation network takes approximately 10 h on a workstation with an Intel Xeon Silver 4212R CPU and an NVIDIA RTX 3090 GPU. While this setup requires substantial investment, it is a one-time process that enables streamlined operations thereafter. In the routine usage phase, our method delivers enhanced efficiency through simplified workflows; users benefit from pre-trained networks, single-round staining with optimized protein pairs, and reduced imaging time compared to traditional cyclic approaches. This streamlined approach eliminates the need for multiple rounds of staining and imaging cycles, making our method increasingly cost-effective with each subsequent use. This efficiency compounds with each use, providing significant time and cost advantages over conventional methods.

Compared to spectral imaging and unmixing, which also allows for the imaging of multiple proteins in a single staining and imaging round, the advantage of SEPARATE is clear. In spectral imaging, the number of images that need to be acquired should be equal to or greater than the number of proteins being imaged. Consequently, as the number of proteins to be imaged increases, images need to be acquired from a narrower bandwidth, which reduces the signal intensities and signal-to-noise ratio. In contrast, in SEPARATE, the number of images that need to be acquired is half the number of proteins, allowing for the acquisition of images from a larger bandwidth. This results in a much higher signal-to-noise ratio, which is critical for imaging proteins with low expression levels.

Although our tissue sections are 150 μm thick as described in the Methods section, we utilized approximately 25 μm depth for imaging due to the limited signal-to-noise ratio in deeper regions caused by antibody penetration constraints. This is standard practice in immunofluorescence imaging, where effective imaging depths are typically less than the total slice thickness. Supplementary Fig. 25 shows that unmixing performance improves with increasing z-stack depth, suggesting that deeper tissue imaging through optimized staining protocols would yield even better results. Since staining occurs bidirectionally from both surfaces of the slice, our method would be readily applicable to thinner sections, such as 50 μm slices, where complete staining penetration requires only 25 μm from each surface.

While our method represents a significant advancement, it comes with several important limitations and areas for future improvement. The main limitation of our technique is that it assumes the expression patterns and levels acquired from singly stained specimens are identical to those of doubly stained specimens. This assumption holds only when antibodies bind to their targets in the same way when applied separately or together, meaning there is no significant crowding effect or antibody crosstalk. For crowding effects, studies report that antibody crowding is severe when target proteins are in close proximity[22]. The antibodies used in this study primarily target cellular markers or proteins expressed in different organelles, where crowding effects are likely minimal or negligible. However, when applying our technique to closely localized targets, such as synaptic receptor proteins within the same synapse or proteins expressed in the same organelle, careful studies on antibody crowding are necessary. Regarding antibody crosstalk, the use of validated antibodies is essential. Antibodies available today undergo more stringent validation[23], but even so, careful evaluation of potential crosstalk is necessary when designing antibody panels for SEPARATE. This may involve additional testing to confirm that antibody combinations do not alter binding specificity or signal interpretation.

Another important consideration relates to the feature extraction methodology. SEPARATE does not explicitly incorporate information about the actual spatial overlap between proteins, as it is designed to determine optimal protein pairings without imaging numerous protein combinations. Common background characteristics can function as important features rather than noise, serving as signals that indicate potential difficulty in unmixing. This feature-based classification approach remains strategically valid for predicting unmixing challenges when direct spatial co-localization information is unknown. Although not currently considered in our framework, incorporating prior knowledge about sub-cellular localization such as which organelle a protein is expressed in could potentially yield more reliable feature vectors even without direct imaging data.

Furthermore, cell density influences feature extraction, as low-density regions provide limited structural cues for learning distinctive representations. This presents a particular challenge for proteins with similar expression patterns, which become harder to distinguish when fewer features are present. SEPARATE addresses this limitation through a random cropping strategy that generates patches with varying ratios of background and cellular regions. This approach allows the model to learn from diverse spatial contexts and incorporate background characteristics as meaningful features, thereby partially compensating for limited information in low-density regions and enabling robust feature extraction across heterogeneous tissue conditions.

Like other deep learning-based biomedical image analysis methods, SEPARATE faces challenges in generalizing from stereotypical to non-stereotypical tissue patterns. This inherent limitation reflects the fundamental machine learning principle that networks require diverse training examples to handle varied scenarios effectively. Consequently, applying SEPARATE to highly heterogeneous samples, such as tumor biopsies or developmental tissues, would require careful consideration of training data diversity to ensure robust performance. However, our validation results across different experimental conditions suggest that the method can adapt to various biological contexts when provided with appropriate training examples.

Additionally, another significant challenge emerges when two proteins show broad, diffuse characteristics in their overlapping regions despite displaying distinct spatial patterns elsewhere. While SEPARATE

incorporates background characteristics into feature learning and generally performs well across entire tissues, in these specific mixed-pattern cases, the network may give more weight to the distinct protein-specific patterns during feature extraction. Consequently, unmixing challenges may arise in overlapping regions where diffuse signals predominate, as the actual accuracy could be compromised when background patterns are prioritized less than the distinctive features elsewhere in the tissue. For such cases, pre-processing steps such as autofluorescence removal[24] or background subtraction can be applied to reduce background interference and enhance unmixing performance. Another potential challenge is the relative brightness between protein pairs; however, SEPARATE maintained robust performance with SSIM values consistently above 0.7 even when varying signal ratios (Supplementary Fig. 28). Beyond this current limitation, future research could explore expanding our method to label more than two proteins with a single fluorophore.

Despite these technical challenges, SEPARATE offers several promising opportunities for advancement and practical applications. While SEPARATE currently involves single-round imaging, it can be readily integrated with iterative staining protocols, where common nuclear stains such as DAPI in the 405 channel can serve as fiducial markers for image registration across multiple rounds. This integration capability enables SEPARATE to be combined with existing multiplexing methods such as PICASSO[18], thereby expanding the palette for multiplexed imaging and further enhancing its utility in protein multiplexing applications.

In terms of imaging compatibility, we validated SEPARATE using spinning-disk confocal microscopy images, but our method could potentially be extended to other imaging modalities such as widefield or light sheet microscopy. However, the unmixing performance might be affected by the presence of out-of-focus light inherent in these systems. This background signal could potentially reduce the accuracy of signal unmixing compared to confocal microscopy, although the aforementioned preprocessing steps might help mitigate these effects.

Among these potential applications, SEPARATE shows promise for diverse applications, particularly in imaging large centimeter-thick cleared specimens. Since staining such specimens typically requires several days, SEPARATE's ability to image multiple proteins using a single fluorophore in one round can effectively reduce the number of required imaging cycles by half, significantly streamlining the process.

## Methods

### Materials
Details of all chemicals and antibodies used in this study are provided in Table 1 and Supplementary Table 1.

### Mouse brain perfusion and slicing
We have complied with all relevant ethical regulations for animal use. All procedures involving animals were approved by the Korea Advanced Institute of Science and Technology Institutional Animal Care and Use Committee (KAIST-IACUC; protocol KA2019-64, KA2025-063). C57BL/6J male mice aged 4–8 weeks were maintained in ventilated cages under standardized conditions, including a 12-h light/dark cycle, temperatures ranging from 20 to 24 °C, and 40–60% humidity. Before perfusion, the mice were anesthetized with isoflurane, followed by transcardiac perfusion with ice-cold 4% paraformaldehyde (PFA) in 1× phosphate-buffered saline (PBS). The harvested brains were subsequently immersed in the same PFA solution at 4 °C for 2 h. Afterward, the brains were sliced into 150-μm sections using a Leica VT1000S vibratome. The slices were then stored in 0.1 M glycine and 0.01% sodium azide in 1× PBS at 4 °C until further analysis.

### Conjugation of Fc-specific Fab fragment antibodies with fluorophores
For the preparation of fluorophore-labeled antibodies, 10 μL of 1 M sodium bicarbonate (pH 8.3) and a ninefold molar excess of succinimidyl ester-fluorophore stock relative to the molar amount of the antibody in dimethyl sulfoxide were added to 90 μL of unconjugated antibody solution. The fluorophore-antibody solutions were incubated at RT for 1 h in the dark. To purify the labeled antibodies, NAP-5 gel filtration columns were used, and equilibrated three times with 3 mL of 1× PBS. Then, 100 μL of the reacted solutions were loaded into the columns. The eluates containing fluorophore-conjugated antibodies were collected after loading 900 μL of 1× PBS. These eluates were concentrated using centrifugal filters with a molecular weight cutoff of 30,000 and adjusted to a final volume of 100 μL with 1× PBS.

### Reaction for preformed antibody complexes and staining
Preformed antibody complexes were prepared following the original primary antibody-Fab complex formation protocol[25]. Specifically, 1× PBS, a solution containing a fluorophore-conjugated Fab fragment antibody, and a solution of primary antibody were mixed in volumes of 10, 2, and 1 μL, respectively. This volume ratio corresponds to a molar ratio of Fab fragment antibody to primary antibody of 6:1 for a primary antibody concentration of 1 mg/mL. The mixture was then incubated for 10 minutes at RT in the dark. Then, 187 μL of blocking buffer (5% normal rabbit serum, 0.1% Triton X-100, 1× PBS) was added to the solution and incubated for 10 minutes at RT in the dark with gentle shaking. For staining, an additional blocking buffer was added to make a final primary antibody dilution ratio of 1:1000. For staining, brain slices were incubated in the diluted antibody solution at 4 °C for overnight in the dark.

### Demonstration of six-plex volumetric multiplexing
We experimentally demonstrated the six-plex volumetric protein imaging with selected antibodies: doublecortin, GFAP, lamin B1, NeuN, nucleolin, and PV. Each pair of antibodies was labeled with one of three fluorescent dyes; CF 488 A, CF 568, and CF 633 in the form of preformed antibody complexes. Each laser channel visualized a pair of NeuN and GFAP (488-nm laser channel), lamin B1 and PV (561-nm laser channel), and doublecortin and nucleolin (647-nm laser channel), respectively. The ratio

## Table 1 | The antibodies used in this study

|  | Name | Host | Vendor | Catalog# | Clone | Dilution |
|---|---|---|---|---|---|---|
| 1 | Calbindin 2 | Rabbit | ATLAS | HPA007305 | Polyclonal | 1:1000 |
| 2 | Calnexin | Rabbit | Abcam | ab22595 | Polyclonal | 1:1000 |
| 3 | Doublecortin | Rabbit | Abcam | ab18723 | Polyclonal | 1:1000 |
| 4 | GFAP | Rabbit | ATLAS | HPA056030 | Polyclonal | 1:1000 |
| 5 | Lamin B1 | Rabbit | Abcam | ab16048 | Polyclonal | 1:1000 |
| 6 | MAP2 | Rabbit | Abcam | ab32454 | Polyclonal | 1:1000 |
| 7 | NeuN | Rabbit | Millipore | ABN78 | Polyclonal | 1:1000 |
| 8 | Nucleolin | Rabbit | Abcam | ab22758 | Polyclonal | 1:1000 |
| 9 | PV | Rabbit | Novus Bio | NB120-11427 | Polyclonal | 1:1000 |
| 10 | S100B | Rabbit | Abcam | ab52642 | Monoclonal | 1:1000 |

of paired primary antibodies in preformed antibody complexes was adjusted according to the manufacturer's instructions.

## Brain sample preparation

For permeabilization and blocking, tissues were incubated in a blocking buffer (5% normal rabbit serum, 0.1% Triton X-100, 1× PBS) at room temperature (RT) for 1 h. The tissues were then stained with a preformed antibody complex mixture at 4 °C overnight and then washed three times with the 1× PBS with Triton X-100 at RT for 30 min.

## Fluorescent imaging

We performed all the imaging experiments using Andor DragonFly 200 spinning-disk confocal microscope equipped with four excitation lasers (405, 488, 561, and 647 nm). Fluorescent images in this study were acquired with water immersion ×60 objective with 1.00 numerical aperture (NA). Images were acquired with a pixel size of 0.1 μm × 0.1 μm, with no averaging steps performed during image acquisition. Images were binned by a factor of two when necessary. For $z$-stack imaging, we used a spacing of 0.5 or 1 μm between optical sections. While our tissue sections are 150 μm thick, imaging was performed to a depth of approximately 25 μm due to antibody penetration limitations and signal-to-noise considerations in deeper tissue regions.

## Experimental dataset for SEPARATE

In this study, we collected three distinct datasets: (1) a feature extraction network dataset, (2) a protein separation network dataset, and (3) a demonstration dataset. Each dataset was obtained from a different animal, with all imaging performed within the DG region. Within each dataset, multiple brain slices from one or two animals were processed together. For each dataset, we acquired multiple $z$-stack images, where each $z$-stack was captured from either a different field of view or a different brain slice. Each $z$-stack image, which corresponds to one imaging session, has dimensions of $1024 \ (x) \times 1024 \ (y) \times 26 \ (z)$ pixels, corresponding to a volume of $204.8 \, \mu m \times 204.8 \, \mu m \times 25 \, \mu m$.

The feature extraction network dataset consists of four $z$-stack images for each protein, acquired from different fields of view within the same anatomical region from one animal. We utilized three $z$-stacks for each protein as training data and one $z$-stack for test data to calculate feature-based distances and identify the optimal grouping. The protein separation network dataset includes four $z$-stack images for each protein pair, acquired from different fields of view within the same anatomical region from two independent animals. For each protein pair, three $z$-stacks were used to train the protein separation network, and one $z$-stack was used to evaluate the unmixing performance and analysis the relationship between the unmixing performance and the feature-based distance. The demonstration dataset comprises ten $z$-stack images, each from different fields of view within two independent animals.

## Feature extraction network architecture

The feature extraction network was designed to capture unique features of the spatial expression pattern of proteins. The feature extraction network consists of five consecutive $3(x) \times 3(y)$ 2D convolutional layers, each followed by a Rectified Linear Unit (ReLU) activation and a $2(x) \times 2(y)$ max pooling operation. The convolutional layers progressively increase in the number of channels—8, 12, 16, 32, and 48, respectively. The output of the last convolutional layer is flattened into a one-dimensional vector and processed through a linear layer, resulting in a final feature vector with a dimensionality of 128.

## Protein separation network architecture

The protein separation network was designed to unmix the signals of paired proteins by leveraging their 3-dimensional (3D) spatial expression patterns. The protein separation network employs 3D U-Net[26] architecture, which comprises a 3D encoder and a 3D decoder connected by skip connections to preserve spatial hierarchies from the encoder to the decoder. The encoder includes $3(x) \times 3(y) \times 3(z)$ 3D convolutional layers, each followed by a ReLU and a max pooling operation. Max pooling is performed using a kernel size of either $2(x) \times 2(y) \times 1(z)$ or $2(x) \times 2(y) \times 2(z)$. The decoder includes 3D transposed convolutional layers for upsampling, each followed by a $3(x) \times 3(y) \times 3(z)$ 3D convolutional layer and a ReLU. The encoder and decoder consist of five 3D convolutional layers with increasing or decreasing the number of channels—8, 12, 16, 32, and 48. The skip connections between the encoder and decoder concatenate feature maps in the channel dimension to link low- and high-level features. At the end of the decoder, $1 \times 1$ convolutional layers are applied to obtain a two-channel output image, each channel corresponding to the unmixed signal of an individual protein. An individual protein separation network is trained for each pair of proteins.

## Training the feature extraction network

The feature extraction network was trained to extract the feature vector of the input protein image, in a contrastive learning manner to minimize the average distance between feature vectors of the same protein and maximize the average distance between feature vectors of different proteins. Each protein image was preprocessed by normalizing it through linear scaling of the 10th percentile value to 0 and the 99th percentile value to 1. Patches of size $256(x) \times 256(y)$ were extracted from the input image by randomly cropping and augmented by random rotations in integer multiples of 90° and flips in the $X$ and $Y$ directions to enhance the robustness of the network during training. The network was trained using a batch size of 512 and optimized with the Adam optimizer[27] at a learning rate of $3 \times 10^{-3}$ without weight decay. without weight decay. The training lasted for 100 epochs, totaling approximately 1 h. The feature extraction network was implemented using PyTorch[28] and trained on a workstation with an Intel Xeon Silver 4212R CPU and an NVIDIA RTX 3090 GPU.

## Training the protein separation network

The protein separation network was trained to unmix the signal of two proteins by minimizing the pixel-wise L1 and L2 loss between the output and the target ground truth images. We utilized a synthetic input image to train the protein separation network instead of attempting to match the relative brightness in the training and test data by optimizing the antibody concentration. First, each image stack of two paired proteins in an identical field of view was normalized by linearly scaling the 10th percentile value to 0 and the 99th percentile value to 1. Thereafter, the synthetic input image was generated by linearly combining two paired protein images after normalization. We used a minimum coefficient of 0.01 and ensured that the sum of the two coefficients used for image blending fell between 0.8 and 1.2 for generating synthetic images. This training process incorporates brightness ratios up to approximately 100-fold between proteins. For the target image, values below 0 in the paired protein images used for synthetic input image generation were truncated to remove the background. Similar to the feature extraction network, patches of size $512(x) \times 512(y) \times 16(z)$ were extracted from the input image stack by randomly cropping and augmented by random rotations in integer multiples of 90° and flips in the X and Y directions. The network was trained using a batch size of 512 and optimized with Adam optimizer at a learning rate of $3 \times 10^{-4}$ without weight decay. The training lasted for 10,000 epochs, totaling approximately 10 h. The protein separation networks were implemented using PyTorch and trained on a workstation with an Intel Xeon Silver 4212R CPU and an NVIDIA RTX 3090 GPU.

## Spatial expression pattern guided protein pairing

To identify the optimal group of protein pairs, we first calculated the distance between clusters formed by extracted feature vectors of each protein. The feature vectors of individual protein images were extracted from the feature extraction network, and the average linkage distance was used to measure the distance between protein clusters in the feature space to infer the distinction between the spatial expression patterns of proteins. The average linkage distance was calculated as the average of all pairwise Euclidean distances between points in the respective clusters.

These feature-based distances served as a precursor for predicting the unmixing performance between protein pairs. Based on these distances, we formulated an optimization problem to identify the optimal protein grouping $G_{opt}$ that maximizes the minimum distance between paired proteins:

$$G_{opt} = \underset{G}{\text{argmax}} \min(d_1, d_2, \ldots, d_n),$$

where G represents the possible grouping of $2n$ proteins into $n$ pairs, and the $d_i$ denotes the feature-based distance between the two proteins in the $i$-th pair ($i = 1, 2, \ldots n$). This optimization criterion, by maximizing the minimum distance between any paired proteins, ensures the most challenging protein pair still maintains sufficient unmixing, thereby guaranteeing consistent and robust unmixing performance across all protein pairs.

For visualization purposes, we employed the t-distributed stochastic neighbor embedding (t-SNE), a dimensionality reduction technique which maps high-dimensional feature vectors into a lower-dimensional space while preserving the relative distances between protein clusters.

### Performance metrics

Structural similarity index measure[28] (SSIM) and Pearson correlation coefficient (PCC) were used as metrics to evaluate the performance of signal unmixing. SSIM quantifies the structural similarity between unmixed image and ground-truth image by measuring luminance, contrast, and structure information, and provide a value between $-1$ and 1, where 1 indicates perfect similarity. PCC quantifies the linear correlation between unmixed image and ground-truth image. PCC values also range from $-1$ to 1, where 1 indicates perfect positive correlation, 0 indicates no correlation, and $-1$ indicates perfect negative correlation. For two protein $A$ and $B$, unmixing performance was calculated by comparing unmixed images $x_A$ and $x_B$ with their corresponding ground-truth images $y_A$ and $y_B$, respectively. These individual performance metrics were then averaged to obtain the final value.

To account for cases where two proteins intrinsically have high spatial similarity, we introduced cross-similarity and the resulting normalized performance metrics. The cross-similarity is defined as SSIM or PCC between ground-truth images of two proteins $y_A$ and $y_B$, and the normalized performance was calculated using cross-similarity, employing the formula:

$$\text{normalized SSIM}(x_A, x_B, y_A, y_B) = \frac{\text{SSIM}(x_A, y_A) + \text{SSIM}(x_B, y_B)}{2(1 + \text{SSIM}(y_A, y_B))},$$

$$\text{normalized PCC}(x_A, x_B, y_A, y_B) = \frac{\text{PCC}(x_A, y_A) + \text{PCC}(x_B, y_B)}{2(1 + \text{PCC}(y_A, y_B))},$$

to avoid large deviation from near-zero denominators.

### Self-supervised image denoising

We utilized a self-supervised image denoising method[29] as a data pre-processing step before training the feature extraction network and protein separation networks. This method employs a 2D U-Net combined with a blind spot network. The input $z$-stack data, excluding the center $z$-plane, are concatenated along the channel dimension and processed through the 2D U-Net to leverage the information from adjacent $z$-planes. The output of the 2D U-Net and the center $z$-plane of the input data are concatenated along the channel dimension and passed to the blind spot network, which has a receptive field set to zero at its center. The output of the 2D U-Net and the output of the blind spot network are concatenated along the channel dimension and fed into $1 \times 1$ convolutional layers to predict the denoised image. The network was trained to minimize the pixel-wise L1 and L2 loss between the input and the output, and optimized with Adam optimizer at a learning rate of $5 \times 10^{-4}$ without weight decay.

### Statistics and reproducibility

Statistical analyses were conducted using MATLAB R2023a (v9.14, Math-Works Inc.). A two-sided paired-sample $t$ test was performed, as described in Supplementary Fig. 17. Exact p-values for all tests are reported in Supplementary Data 2. The sample size ($n$) refers to images acquired from different fields of view within the same anatomical region from one animal, and does not represent biological replicates. Details are provided in the figure legends.

### Reporting summary

Further information on research design is available in the Nature Portfolio Reporting Summary linked to this article.

### Data availability

The data generated in this study are available from the corresponding authors upon reasonable request. The public human cell dataset from Allen Institute for Cell Science can be downloaded from https://downloads.allencell.org/publication-data/label-free-prediction/index.html. Source data underlying all graphs are provided as Supplementary Data 1 and 2.

### Code availability

Code for Pytorch implementation is available online at the GitHub repository (https://github.com/NICALab/SEPARATE) and has been deposited on Figshare (https://doi.org/10.6084/m9.figshare.29148770.v1[30]).

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

## Acknowledgements

This research was supported by National Research Foundation of Korea (NRF) (RS-2023-00209473, RS-2023-00264409, NRF-2019R1F1A1063145), Korea Basic Science Institute (National research Facilities and Equipment Center) grant funded by the Ministry of Science and ICT (RS-2024-00401676), Bio & Medical Technology Development Program through the NRF funded by the Ministry of Science and ICT (RS-2021-NR056586), the BK21 plus program through the NRF funded by the Ministry of Education of Korea, and the Korea Medical Device Development Fund grant funded by the Korean government (1711137947, KMDF_PR_20200901_0027).

## Author contributions

G.K. and M.E. designed a protein pairing and signal unmixing algorithm. G.K. and M.E. designed and performed experiments and analyzed data. H.S. and H.K. performed immunofluorescence imaging of mouse brain slices. G.K., J.-B.C. and Y.-G.Y. wrote the manuscript with input from all authors. J.-B.C. and Y.-G.Y. conceived and led this work.

## Competing interests
The authors declare the following competing interests: G.K., H.S., J.-B.C., Y.-G.Y., H.K., and M.E. are co-inventors on a patent owned by KAIST covering SEPARATE. Y.-G.Y. and J.-B.C. are the co-founders, shareholders, and employers of a company specializing in various imaging services; this research was conducted independently and is not affiliated with the company.
