## [Transparent Peer Review file · Communications Biology]

Doubling multiplexed imaging capability via spatial expression pattern-guided protein pairing and computational unmixing

Corresponding Author: Professor Young-Gyu Yoon

Version 0:

Reviewer comments:

Reviewer #2

(Remarks to the Author)

Summary

Kim et al. present an approach where pairs of proteins can be assigned to single detection fluorophores for imaging based on distinct spatial expression patterns based on a convolutional neural network (CNN) for feature extraction and can then be separated by another CNN. Theoretically this allows us of N fluorophores to visualize 2N proteins. They apply this approach for volumetric imaging of 6 proteins using 3 fluorescent channels.

The idea of using one fluorophore/fluorescent channel for imaging of 2 spatial non-overlapping targets is rather intuitive and not very novel in essence. But here the authors actually aim to provide a more systematic and automated approach to quantitatively define which protein pairs are "spectrally non-overlapping" and are suitable to be imaged with one fluorophore. This idea is very attractive and interesting, however the implementation is not clearly described and some important considerations seem to be overlooked, including (i) the balance of signal distributions for the target pairs, (ii) how the feature-based distances may be biased by features that cannot be captured by the unmixing model when the same fluorophore and imaging settings are used, (iii) how the use of training data from different samples/regions may affect the feature extraction. Many relevant technical details are also missing and limitations are not discussed. This makes it hard to evaluate the performance, general applicability, robustness and versatility of the method. As is, the paper does not present SEPARATE in a sufficiently scrutinised manner, despite being primarily a methods paper.

Major Comments

1. It is hard to understand what the imaging context where this method would perform best is. In principle, one could imagine two cases: (1) higher-resolution imaging where subcellular distributions of target proteins are differential and can be leveraged (2) lower resolution imaging for tissues where differential localization of marker proteins (such as cell type markers) can be leveraged. Of course there could be a third case that is a hybrid of these two. In each case, there would be differences in the imaging conditions that would work best for separation, and limitations in useability of the presented approach. In the manuscript authors do not really define the application space, so it get hard to understand:

- Why are the given proteins selected?
- What are ideal imaging conditions, and how these may affect the results?
- What are the limitations of the approach?

2. What are the criteria for optimal groupings? It is unclear how the particular groupings are done after generating the pairwise distance matrix for Fig. 2 (what are the comparative maximized minimum distances for the 2 groups). This needs to be better described.

3. The manuscript does not mention or discuss the applicability of their approach when it comes to incorporation general nuclear and membrane stainings - two common markers used for basic feature segmentation of cells and tissue. These two labels would be present in all the cells and overlap with other markers in the panel and therefore present a major challenge to SEPARATE unmixing strategy.

4. A similar question could be asked for membrane based cell type markers, which may not normally overlap in the same cell, but might have partial overlaps in the tissue when these cells are next to each other (and the abundance of these incidents would vary with the region of the tissue). An example case for this would be immune cells in a tonsil sample. How would the approach deal with these kinds of situations?

5. If the reference images for training of the feature extraction network were coming from the same sample and field of view, one could very easily expect a good predictability from the distance matrix. However, when images are from different cells and regions (which seems to be the case here based on Fig. 2 and Supplementary Fig. S1), it is unclear to me how randomness of the region selection would affect the extracted features, and how big the dataset needs to be to compensate for heterogeneity in signal distribution especially in tissue samples like the brain where cell type abundances and noise can vary across the tissue, and especially when there is no cell segmentation etc.

6. If utilizing singleplex images from different samples (not obtained from the same tissue slice or cells) is sufficient to calculate reliable pairwise distance, can authors utilize an existing data set (such as human protein atlas), where reference images are available? If not, what are the limitations for this?

7. Suppl Fig. 1: I'd be curious to see patch examples for the non-overlapping parts of the same clusters (for example nucleolin/NeuN) - how come there are clearly separated parts of these clusters although images suggest that they would have very similar staining patterns overall.

8. The above question also points to a critical weakness of the method. It would be very hard to unmix proteins that are spatially overlapping in the same cell when a single fluorophore is used. The feature vector may find some high distance between them if the weight of the features is not adjusted to avoid this scenario because some features may just not be accessible for unmixing when the same fluorophores are used. Similarly other features like absolute signal intensity/ target abundance etc. may not be properly represented in the feature matrix because training images are not taken under comparable conditions (same fluorophore, same imaging settings etc.) - (I presume this is the case because no information is provided about how these stainings and imaging was done, which has to be included in the Methods. Under these circumstances it is really not clear to me how good the feature-based distance can actually accurately capture the pairs for best unmixing success.

9. Could the authors please explain the t-SNE in Fig. 2 better? Is each dot one patch of one image/FOV? In the Reporting Summary document, they mention three datasets were used for training and 1 dataset was used for feature extraction. What exactly is one dataset and how big they need to be for reproducible feature extraction? If it was a single dataset that was utilized for analysis, how different would the data in Fig. 2b-c look for an independent repetition (same markers different sample/ imaging region)?

10. Similar to the point above, what are n (sample sizes) and the error bars for different graphs? Is almost all of the main data that went into the manuscript generated by imaging just one field of view for each target, and z positions are used for getting the sample number to have error bars? This is not clear for Fig. 3e-f. Unless the z -step sizes are really big, distributions from one position to the next might be quite similar at least for some targets, which would be an argument to not treat them as different samples for statistics.

11. Even for markers that are spatially distant, the unmixing performance is not great. For example in the network output for PV one could clearly see the remnants of Lamin B1 which is not visible in the groundtruths in Fig. 3. Based on this, how is Fig. 4 result should be evaluated? At the lower right corner of the Fig. 4f I could clearly see red nuclei with bright nucleoli - these are assigned to Doublecortin, however doublecortin is found in the cytoplasm, and the unmixing probably failed by wrong assigning nucleolin to doublecortin. Interestingly, this is the pair, where SSIM was highest.

12. The SSIM data is the main benchmark in the paper, however it could be seen that even for the cases of pairs with low distances SSIM values are quite high (over >0.9), suggesting this may not be the best measure. It is obvious that SSIM would not be very sensitive for the actual unmixing performance when highly overlapping distributions (such as nucleolin/NeuN) are being evaluated, as the NeuN similarity to nucleolin groundtruth and vice versa would already be very high. Should this at least not be also plotted or used for normalization.

13. How do different factors, such as resolution, autofluorescence and noise of the reference images affect the assignment of pairs? Authors can at least process the same data (for example by binning) to see how the performance is getting affected. Authors mention in the methods that they apply self-supervised denoising to the images before using them for training, how necessary is this? And is it critical for getting good performance?

14. When two different targets are combined into one imaging channel, they need to have similar brightness so that one acquisition setting could give sufficient but not saturating signal for both targets. This would be a critical consideration for how to make pairs. This is not mentioned or discussed at all, and definitely needs to be addressed. And if the networks were somehow accounting for this, it would be actually very useful.

15. Line 171-175: Authors discuss the requirement of 3D images for training. What thickness is relevant (not mentioned in the manuscript). Would 2D projections of 3D stacks not be sufficient for training? Projections usually capture the overall signal morphology (such as fibrous versus spotty). Related to this and imaging process, many relevant technical details are omitted. According to the Methods, sections are 150 μm thick, but based on the images it seems like maybe only the upper layers were imaged. It is unclear what the actual thickness is and what kind of z -step size is used.

16. The authors are going with a relatively small set of markers right now. How does the method scale when more protein markers are included?

17. Methods section: For prelabeling of antibodies with Fabs, the Fab-antibody molar ratio is critical. In the methods section only a volumetric ratio is given, which is not informative. When the antibody-Fab binding is not saturated there is a big risk of crosstalk, which would affect the results for this kind of a study (especially here all the antibodies are from one source species). Additionally, antibodies and clones (vendors, source species, dilutions) need to be also reported in the main text.

Minor Comments

- Intro Line 44: Authors state: "However, as staining typically takes more than 12 hours and can extend to several days for thick specimens²³, performing multi-round staining and imaging could significantly prolong the entire process."

Imaging cycles could be much shorter (down to 15-30 min with thin sections and a couple of hours for thicker samples) for methods that do not require antibody relabeling, such as DNA oligo barcoding.

What are the coefficients used for generation of synthetic images?

- Presenting single channel microscopy data in grayscale would be more ideal (Fig. 2a; 3a-b; S1b-f).

- Use of different color combinations for every image in Fig. 3 overlays are not helping (one color combination, such as green/magenta would be sufficient) and with the chosen color schemes it is hard to see the actual signal overlap (like two blues in NeuN/nucleolin).

- Plots in 3e & f are considered as main results and are too small to properly see. Fig. 3 should show input/output for all targets.

- Developmental stage / age of sacrificed mice is missing.

- Researchers testing the SEPARATE approach would benefit from parameters on how to select pairs in complex panels (i.e. what is the minimum distance that is still acceptable between targets).

- Would there be any limitations in the fluorophore selection?

- How does the computational unmixing deal with auto-fluorescence that can range from general tissue background (FFPE tissue blocks) to bright signals in select cells (blood vessels)?

- Fig4 c,d,e - protein names are not written for the respective images.

- There is no mention of how the fiducials (bright, defined landmarks in a single color / dye commonly used in iterative staining protocols) could be utilized in parallel to the SEPARATE.

- It would be interesting to see the performance of the computational unmixing in a subcellular panel with defined markers at higher resolution where artifacts can be more easily observed.

Fig2c would be better to see the actual numbers for the colormap.

Line 329-241: "Protein pairs with larger feature-based distances were more accurately unmixed, with a Pearson correlation of 0.8728 between the feature-based distance and the average SSIM. Scale bar = 30 μ m in a-d." - Is the given PCC for the pair with highest SIM? PCC values should be given for all conditions.

Reviewer #3

(Remarks to the Author)

In this manuscript Kim G. et al. describe SEPARATE, a method for multiplexed fluorescent imaging aimed at halving the time required to detect up to 10 protein targets. The authors achieve this by identifying proteins pairs that could be detected simultaneously with antibodies labelled with the same fluorophores, and their staining patterns later deconvolved using a feature extraction network previously trained. 3D volumetric imaging of a thick brain section is highlighted as validation of the method.

The idea of reducing the processing requirements of multiplexed imaging is indeed intriguing and valuable as there is a great need of original solutions aimed at mining datasets beyond traditional approaches. Yet, the solution proposed by the authors appears quite burdensome and of difficult implementation, especially for small plexes where other solutions are already available. The panel set up and training of the feature extracting network requires a significant commitment upfront (time and resources), which could only be justified for large cohorts of samples to be processed (i.e. large atlases) but not for more routinely used sample sizes. Additionally, the performance of the method is only shown for non-pathological samples in a tissue (brain) with very stereotypical expression patterns. It is not clear if SEPARATE would be applicable to the study of more complex sample cohorts where sourcing appropriate training data would be not trivial. This might be the case of patient cohort or embryonic development when expression patterns change as a result of disease or development for example, or in non-stereotypical samples such as tumour biopsies.

More should be added on the size and specs of the training dataset. From the information available in the Reporting Summary it appears to be extremely limited and arbitrary.

The authors state that SEPARATE fills a void in the field by halving the processing time of cyclic imaging solutions, which they say range from 12hrs up to days long processing times. However, there are both academic protocols (e.g. HIFI, Watson S. et al, 2024 Nat Comm, <https://www.nature.com/articles/s41467-024-47185-9#Sec12>) and commercial solutions (Lunaphore COMET) with significantly shorter processing requirements (5-6 hrs/cycle for HIFI, 5hrs total for Lunaphore COMET).

Moreover, it is worth noting the existence of several commercial solutions able to achieve similar results. In example, Thermo Scientific has recently announced the release a new commercial spatial imaging solution (EVOS S1000) which used spectral deconvolution of optimised fluorophores to simultaneously detect up to 8 different antibody targets plus DAPI. As a general note, the method section is devoid of many key details (albeit some present in the Summary Report) that would impact the reproducibility of the method. Similarly, it would be appreciated if the authors could release the code used for this

manuscript.

More detailed comments below:

1. line 37. Imaging Mass Cytometry is not based on cyclical fluorescent imaging, yet is similarly adopted.
2. Line 107. The authors use synthetic images to generate training data for the feature extraction network. This is by linearly superimposing individual single staining images, circumventing the need to perform antibody optimization. However, this training dataset is methodologically different from the input data where the staining would have physically occurred simultaneously, and therefore does not take into account the behaviour of the antibodies when mixed together, for example competition, crowding, aspecificity or colocalization. It would be appreciated if the authors could comment on this or present data to address this issue.
3. Line 118. No mention is made in the text in reference to how many images are used to train the networks. There is some limited detail in the Reporting Summary, which should be added in the main text and more in detail in the methods. Additionally, the number of images stated is much more limited than other papers using feature extraction approaches on imaging datasets. Could the authors comment?
4. The SSIM metric chosen by the authors quantifies across the entire field of view. It would be more appropriate if the authors could also provide a spatially aware metric of their methods to evaluate the accuracy across different cellular or tissue structures, position within the z stack and relative position within the field of view.
5. Line 150. The authors should include in the text the thickness of the sample and details of the imaging. In the current wording "volumetric" could be easily interpreted as applied to larger samples than a tissue section.
6. Line 168. The authors should validate their method on more than a single sample and provide a metric to demonstrate reproducibility, ideally across different tissue types and different staining.
7. Line 180. The authors should increase the sample size and diversity of tissues/antigens in order to comment on the robustness.
8. Line 197. This claim is not fully supported by the data presented as the thickness of the sample shown in the manuscript is significantly smaller (150 μm), and this could impact the performances of the stainings and therefore of the algorithms.
9. Moreover, it would be good if the authors could comment on the portability of SEPARATE to samples imaged with other methods, such as widefield or light sheet microscopy, which are commonly used in other multiplexed imaging settings.
10. Line 198. The authors could include the time required to set up SEPARATE from choosing the optimised antibody pairs to training the feature extraction algorithm and running it on imaging datasets.
11. Line 229 (figure 2c). The scale bar is missing values.
12. Line 234 (Figure 3c, 3d). This would be better substituted with supplementary figure 2. Although the consistency with the tSNE coloring scheme is appreciated, it negatively impacts the readability of the figure, as some of the color pairs cannot be easily discerned (for example in the calbindin 2/calnexin or NeuN/nucleolin). Ideally this should be color-blond friendly.
13. Line 246 (Figure 4c-e). could the authors label images with the proteins displayed similarly to 4a?
14. Line 260 and 278. No details of the antibody used and succinimidyl ester-fluorophores used. Similarly, the authors should state the actually primary antibody concentrations used
15. Line 291. The authors should provide more details on the imaging set up, for example pixel size, presence of any averaging step, spacing of the optical sections in the z-stacks.

Version 1:

Reviewer comments:

Reviewer #1

(Remarks to the Author)

The authors have addressed the comments and concerns that the reviewers made in a detailed manner. They have now also included additional figures in the main and the supplemental that explain more details of the methodology. The limitations have been also added to the main manuscript. There are no further comments from my side.

Reviewer #2

(Remarks to the Author)

The authors have made a significant effort in addressing many of the points raised during the initial review. The inclusion of the Allen Cell dataset, additional figures, and clarifications in the methods section have strengthened the manuscript. The analysis of different factors, such as the impact of resolution and noise of the reference images are also important new insights.

However, several concerns remain that need further clarification, along with some technical and presentation-related improvements to enhance the robustness and clarity of the study.

Major Comments:

1. Interpretation of Allen Cell Dataset Results: The results from the Allen Cell dataset raise concerns regarding the practical usability of the method. Intuitively, non-spatially overlapping markers like DAPI and ZO1 should have the highest separability; however, these markers performed poorly in the feature distance matrix. This discrepancy warrants further investigation and explanation.
2. Reporting SSIM: In Figure 3b, 5d-e and Supplementary Fig. 19, it is unclear which channel was used for calculating the

SSIM values. How is the SSIM calculated for the unmixed versus groundtruth? I'm guessing single-channel images were used for this, and in this case SSIM should have been reported for both of the channels. It is not specified which one we are seeing in panels d and e, but I am guessing it is the SSIM for the markers in the x-axis, and for the cell membrane or DAPI. However, the similarity might be quite different if they quantify the SSIM and PCC for cell membrane or unmixing for each pair, which would actually make it possible to understand how the pairing is affecting the unmixing of the same protein in each case.

If my assumption is wrong and the SSIM is being reported for multi-color images, I do not think this is appropriate.

Additionally, the manuscript utilizes SSIM almost exclusively for evaluating the unmixing performance. SSIM (between unmixed and groundtruth) might not be the best stand alone metric for evaluating the unmixing performance, especially if the aim is seeing how well a marker is separated from the other (which was also partially raised in Point 12 before). In their calculation SSIM values appear quite high for most conditions (above 0.7 in most cases), which suggests it is not very sensitive to critical changes in spatial distribution. Other correlation coefficients like PCC or Mander's are expected to be more suitable in this case to show reliable separation (rather than similarity which also looks at different factors that are not as relevant here), and all of the unmixing performance measurements need to be done separately for both channels that are being unmixed and reported as such. They have now included Supplementary Fig. 19 with PCC calculations, but as noted above it is unclear how these are calculated for pairs and what is being reported. Similarly, not much explanation is given about how cross-SSIM or cross-PCC is calculated. This is also only done for one of the main experiments, and is included only as a supplement.

3. Additionally, the manuscript lacks statistical significance testing for the SSIM values of different protein pairs, which generally appear quite high for most conditions (for example in Fig. R1d-e or Fig. R10). Reporting statistical comparisons would improve the interpretability of these results and help understand the actual improvement between pairs that have a high distance and low distance in the feature-based distance matrices.

There should also always be a negative control case, where similarity/correlation is expected to be naturally low.

For calculating statistics authors refer to 16 patches obtained from the same image as "independent" patches - these are not independent.

4. Boxplot Visualization (for eg. Figures 3b and 5d-e and Supplementary Fig. 19): The feature-based distance is presented as a continuous dotted line, which makes it difficult to see the actual values. Since these distances represent distinct spots, connecting them with lines can be misleading. I recommend using distinct markers without connecting the data points. Also, I would recommend plotting the boxes in the order of decreasing/increasing distances (or SSIM), so one could see the trend more clearly.

Median vs. Average in Line 330: The authors mention using the median in the boxplots but calculate PCC using the average SSIM. This discrepancy should be resolved, and the choice of metric should be clearly justified.

It seems in both figures there are some outliers (which are quite hard to see with the current label). Especially for Fig. 3, which patches are regarded as outliers (since they are all shown in Supplementary Fig. 8 and 9), and how are they justified.

6. The two-color overlays used in many figures (for eg. Supplementary Fig. 3-8) are not helpful to have a reliable visual comparison of unmixed images to ground truth images, these should be presented as single by side single channel grayscale images.

6. Figure R5 clearly shows that the cell density in the image patches is a crucial factor for feature vectors, but the authors do not discuss the implications of this observation. This is an important discussion to understand the limitations of the approach. As such, insufficient sampling of the patches (for example having lots of empty tissue areas included in the input dataset) could yield very different feature vectors.

Minor Comments:

- Supplementary Figure 8 - how local unmixing accuracy was calculated is not explained in the legend.

-Thickness: In the methods section authors state "patches of size $512(x) \times 512(y) \times 16(z)$ were extracted from the input image stack". 16 z sections mean either 8 or 16 μm thick z-stack, which is smaller than what they state in the methods. Could the authors provide additional reasoning which z planes were removed and why the z-stack was reduced from $25\mu\text{m}$ to 8 or 16 μm ?

-Introductory Statement (Line 44): The authors should clarify their statement regarding staining duration, as methods like DNA oligo barcoding allow for much shorter cycles.

-Synthetic Image Representation (Figure R2): Synthetic images should clearly label both markers, as they are mixtures of the respective markers. The manuscript does not specify which color corresponds to which marker. The coefficients used for generating synthetic images should be added to the methods section.

-Dataset Clarification (Figure R4): The figure does not clearly indicate which dataset was used for training and which for testing. Furthermore, it should be stated whether any FOVs were reused in both datasets.

-Grayscale Presentation: Single-channel microscopy images should consistently be presented in grayscale for better interpretability.

-Figure 3 Overlays: The main results in Figure 3 should be zoomed in to improve visibility.

-Feature-Based Distance Thresholds: Quantitative thresholds for acceptable feature-based distances should be provided to assist researchers in applying SEPARATE to their own datasets.

-Figure 4 Labeling: Protein names should be explicitly labeled in all sub-panels.

-Use of Fiducial Markers: The authors suggest using DAPI as a fiducial marker but mention it being in the 488 channel instead of 405 nm. This needs correction.

-Heatmap Values: Numeric values would ideally be added to the heatmap for better interpretability.

-Formula Documentation: Formulas for metrics like cross-SSIM should be clearly documented.

Version 2:

Reviewer comments:

Reviewer #2

(Remarks to the Author)

The authors have addressed all my comments in the last revision, which improved the transparency and accurate presentation of the data and the quality of the discussion. I do not have further concerns.

Response to review

We would like to express our gratitude for the insightful and constructive comments from the reviewers. These comments have helped us to substantially improve our manuscript. We have carefully read and addressed each point raised by the reviewers. Please find our detailed responses to each reviewer's comments below. Changes have been made in the manuscript accordingly.

Below is a brief summary of the major changes in response to the review comments:

1. We have extensively validated SEPARATE across more samples and different tissues/antigens. We tested our method on the public human cell dataset from Allen Institute for Cell Science containing 10 proteins with diverse subcellular localizations. This validation demonstrated robust performance across different species and imaging conditions, with strong correlations (0.8225 and 0.8853) between feature-based distances and unmixing performance (We note that we are not looking for the correlation values that are close to one; these are for checking whether the feature distance is a good enough precursor of the unmixing performance). The results are now included as Figure 5 in the manuscript and Supplementary Figures 14 and 15.
2. We have expanded our methodological details and technical validation:
 - Added comprehensive cross-validation analysis using four independent datasets to demonstrate reproducibility (Supplementary Figure 11)
 - Conducted detailed analysis of how imaging conditions affect performance, including resolution effects (Supplementary Figure 12) and noise impact (Supplementary Figure 13)
 - Provided complete specifications of imaging parameters, antibody details, and experimental protocols in the Methods section and new Supplementary Tables 1 and 2
3. We have addressed the limitations of our method and added extensive new analyses:
 - Evaluated how relative signal intensity between paired proteins affects unmixing performance (Supplementary Figure 20)
 - Analyzed the impact of tissue thickness and imaging depth on performance (Supplementary Figures 16-17)
 - Added detailed discussion of antibody crowding effects and their implications
 - Included comprehensive spatially-aware performance metrics (Supplementary Figures 8 and 9)
4. We have strengthened our performance analysis with multiple new experiments:
 - Added normalized metrics accounting for baseline cross-similarities between proteins (Supplementary Figure 19)
 - Conducted Monte Carlo simulations to evaluate scalability with larger marker sets (Supplementary Figure 18)
 - Performed detailed analysis of unmixing performance across varying experimental conditions
5. We have expanded the discussion of practical implementation considerations:
 - Added detailed protocol for antibody labeling and optimization
 - Included comprehensive guidelines for optimal imaging conditions
 - Provided clear explanations of the setup process and routine application procedures

Reviewer #2 (Remarks to the Author):

Summary

Kim et al. present an approach where pairs of proteins can be assigned to single detection fluorophores for imaging based on distinct spatial expression patterns based on a convolutional neural network (CNN) for feature extraction and can then be separated by another CNN. Theoretically this allows us of N fluorophores to visualize 2N proteins. They apply this approach for volumetric imaging of 6 proteins using 3 fluorescent channels.

The idea of using one fluorophore/fluorescent channel for imaging of 2 spatial non-overlapping targets is rather intuitive and not very novel in essence. But here the authors actually aim to provide a more systematic and automated approach to quantitatively define which protein pairs are "spectrally non-overlapping" and are suitable to be imaged with one fluorophore. This idea is very attractive and interesting, however the implementation is not clearly described and some important considerations seem to be overlooked, including (i) the balance of signal distributions for the target pairs, (ii) how the feature-based distances may be biased by features that cannot be captured by the unmixing model when the same fluorophore and imaging settings are used, (iii) how the use of training data from different samples/regions may affect the feature extraction. Many relevant technical details are also missing and limitations are not discussed. This makes it hard to evaluate the performance, general applicability, robustness and versatility of the method. As is, the paper does not present SEPARATE in a sufficiently scrutinised manner, despite being primarily a methods paper.

We appreciate the summary and the insights provided. In our revised manuscript, we have thoroughly addressed the concerns regarding implementation clarity and methodological considerations. Regarding the implementation clarity, we have enhanced Methods section and Reporting summary with detailed description of our approach. The revised Methods section now provides detailed explanation about the optimal grouping criterion (please see major comment 2), the dataset specifications (please see major comment 9), the image acquisition setup (please see major comment 17 and minor comment regarding the developmental stage and age of sacrificed mice), and the coefficient used for synthetic image generation (please see minor comment).

Furthermore, we have conducted additional experiments to rigorously assess the critical consideration reviewer raised: (i) analysis of relative signal intensity between paired two proteins (please see Figure R8), (ii) assessment of potential biases in feature-based distances arising from imaging parameters (please see Figure R6 and Figure R11), (iii) evaluation of how sample and regional variations impact feature extraction (please see Figure R4 and Figure R7), (iv) demonstration of SEPARATE's versatility across different species (please see Figures R1-3). These comprehensive additions not only address the technical concerns raised but also provide robust validation of SEPARATE's performance and broad applicability.

Major Comments

1. It is hard to understand what the imaging context where this method would perform best is. In principle, one could imagine two cases: (1) higher-resolution imaging where subcellular distributions of target proteins are differential and can be leveraged (2) lower resolution imaging for tissues where differential localization of marker proteins (such as cell type markers) can be leveraged. Of course there could be a third case that is a hybrid of these two. In each case, there would be differences in the imaging conditions that would work best for separation, and limitations in useability of the presented approach. In the manuscript authors do not really define the application space, so it get hard to understand:

- Why are the given proteins selected?
- What are ideal imaging conditions, and how these may affect the results?
- What are the limitations of the approach?

We thank the reviewer for the insightful comment that helps us better frame SEPARATE's application space. SEPARATE is designed for neuroimaging applications, addressing the complex challenges of studying the brain's cellular composition by leveraging distinct spatial expression patterns in tissue. To demonstrate SEPARATE's effectiveness, we selected proteins widely used and imaged in neuroscience studies. These include cell type markers such as PV, GFAP, NeuN, S100B, and doublecortin, as well as proteins associated with specific cellular organelles, including lamin B1, MAP2, calnexin, and nucleolin. This focus is particularly relevant for neuroscience applications, where understanding cellular composition and organization is crucial for mapping brain structure and function.

For optimal performance, SEPARATE requires imaging conditions that can reliably capture tissue-level spatial expression patterns of marker proteins (please see Figure R11, where we analyzed how the performance of SEPARATE is degraded as a function of the image resolution). Consistent noise levels across different protein images (please see Figure R6) and minimal tissue autofluorescence are essential for accurate unmixing. Sample preparation quality and consistent immunostaining are also crucial factors that influence the method's performance, as they directly impact our ability to detect genuine differences in protein expression patterns.

One limitation of the protein pairing procedure arises from the fact that the features are extracted from singly stained images without taking physical overlap between proteins into account. However, what truly matters is avoiding the pairing of proteins indistinguishable by the network – specifically, proteins that exhibit similar spatial expression patterns and substantial spatial overlap. SEPARATE addresses this challenge by maximizing the minimum distance between protein pairs, ensuring paired proteins have distinctive spatial expression patterns. Furthermore, the protein separation network can detect and exploit subtle differences in the proteins' 3-dimensional profiles through its 3-D convolutional layers.

Despite these technical considerations, SEPARATE is particularly advantageous for specimens with a thickness beyond 10 μm , such as 35 μm , offering significant time savings over conventional cyclic immunofluorescence while maintaining multiplexing capability. We have expanded these points in the revised manuscript to provide clearer guidance on the method's optimal application space and limitations.

2. What are the criteria for optimal groupings? It is unclear how the particular groupings are done after generating the pairwise distance matrix for Fig. 2 (what are the comparative maximized minimum distances for the 2 groups). This needs to be better described.

We thank the reviewer for pointing this out. Given $2n$ proteins, the number of possible grouping is $\frac{(2n)!}{2^n n!}$. Our criterion for optimal grouping involves finding the grouping that maximizes the minimum distances between paired proteins. Specifically, we search for the optimal group G_{opt} defined as follows:

$$G_{opt} = \operatorname{argmax}_G \min(d_1, d_2, \dots, d_n),$$

where G represents the possible grouping of $2n$ proteins into n pairs, and the d_i denotes the feature-based distance between the two proteins in the i -th pair ($i = 1, 2, \dots, n$).

Our optimization criterion utilizes a feature-based distance metric that quantifies the expected unmixing performance between protein pairs. By maximizing the minimum distance among paired proteins, we ensure robust unmixing even for the most challenging protein combinations. This approach emphasizes consistent performance across all protein pairs, rather than allowing exceptional unmixing in some pairs at the expense of others. As demonstrated in Figure 2, the optimal grouping, group 1, achieves a minimum distance of 2.0039, while the alternative grouping, group 2, yields a minimum distance of 1.1601.

A detailed description of this grouping criterion is provided in the Methods section, and the specific minimum distance values for both groups are included in the caption of Figure 2.

3. The manuscript does not mention or discuss the applicability of their approach when it comes to incorporation general nuclear and membrane stainings - two common markers used for basic feature segmentation of cells and tissue. These two labels would be present in all the cells and overlap with other markers in the panel and therefore present a major challenge to SEPARATE unmixing strategy.

We acknowledge that general nuclear and membrane staining is present in all cells and often overlaps with multiple other markers. However, our unmixing strategy is specifically designed to identify protein pairs that can be effectively separated using a feature extraction network, by avoiding combinations of proteins with similar spatial expression patterns. In other words, having to unmix two proteins that both stain the membrane is exactly what we can avoid by our optimal grouping approach.

To validate our approach, we tested SEPARATE on a public human cell dataset from Allen Institute for Cell Science¹ containing 10 proteins: alpha-tubulin, desmoplakin, lamin B1, myosin IIB, Sec61 translocon beta subunit (Sec61 β), ST6 beta-galactoside alpha-2,6-sialyltransferase 1 (ST6GAL1), translocase of outer mitochondrial membrane 20 (TOMM20), tight junction protein 1 (ZO-1), cell membrane labeled with CellMask, and DNA labeled with Hoechst (Figure R1a). This dataset provides image stacks where cell membrane and DNA are imaged together with one of the other proteins in the same field of view. Using the feature extraction network, we obtained feature vectors for each protein and visualized them in 2 dimensions using t-SNE (Figure R1b). Based on the pairwise feature-based distances between protein clusters, we determined the optimal grouping that maximized the minimum distance between paired proteins: (alpha-tubulin, DNA), (desmoplakin, ZO-1), (lamin B1, cell membrane), (myosin IIB, ST6GAL1), and (Sec61 β , TOMM20). The optimal pairs are highlighted by dashed boxes in the pairwise feature-based distance matrix (Figure R1c). Notably, these optimal grouping successfully avoided combining proteins with similar spatial expression patterns. Membrane-associated proteins ZO-1 and cell membrane were assigned to different pairs, and cytoskeletal proteins alpha-tubulin and myosin IIB were also assigned to different pairs. Although desmoplakin and ZO-1 are both membrane-associated proteins, they show distinct spatial expression patterns as desmoplakin exhibits punctate patterns at desmosomes while ZO-1 shows fibrous patterns at tight junctions.

We then analyzed the relationship between these feature-based distances and unmixing performance, quantified by SSIM between ground truth and unmixed images. The analysis was performed on two sets of protein pairs: 8 pairs consisting of cell membrane with each of the other proteins (Figure R1d) and 8 pairs consisting of DNA with each of the other proteins (Figure R1e). This analysis showed that protein pairs with larger feature-based distance achieved more accurate unmixing, with a Pearson correlation between the feature-based distance and mean SSIM of 0.8225 for proteins paired with cell membrane and 0.8853 for proteins paired with DNA. These results demonstrate that our feature-based protein pairing strategy effectively predicts unmixing performance also in the public human cell dataset from Allen Institute for Cell Science. Figure R2 and Figure R3 show the unmixing results in channel-wise merge using a green-and-magenta color scheme of proteins paired with cell membrane and DNA, respectively.

This is now added as Figure 5 in the manuscript, and Supplementary Figures 14 and 15 in the supplementary materials.

Figure R1. Validation of SEPARATE on the public human cell dataset from Allen Institute for Cell Science. **a**, Representative images of ten proteins in the public human cell dataset from Allen Institute for Cell Science: alpha-tubulin, desmoplakin, lamin B1, myosin IIB, Sec61 β , ST6GAL1, TOMM20, and ZO-1. **b**, Two-dimensional visualization of feature vectors using t-SNE. Each dot represents a feature vector extracted from a single image patch of protein. **c**, The pairwise feature-based distance matrix computed between clusters of feature vectors for each protein pair and displayed as a heatmap with a red-yellow-blue color scheme, where blue indicates higher and red indicates lower feature-based distances between protein pairs. **d**, Comparison of SSIM and feature-based distances for 8 proteins paired with cell membrane. SSIM between ground truth and unmixed images was calculated for each image stack ($n = 11$ per pair) and displayed as a box-and-whisker plots. The boxes show the interquartile range (IQR) with the median, while whisker extend to 1.5 times the IQR. Individual points represent outliers. Protein pairs with larger feature-based distances showed more accurate unmixing, with a Pearson correlation of 0.8225 between the feature-based distance and mean SSIM across z-stacks. **e**, Comparison of SSIM and feature-based distances for 8 proteins paired with DNA. SSIM between ground truth and unmixed images was calculated for each

image stack ($n = 11$ per pair) and displayed as a box-and-whisker plots. The boxes show the interquartile range (IQR) with the median, while whisker extend to 1.5 times the IQR. Individual points represent outliers. Protein pairs with larger feature-based distances showed more accurate unmixing, with a Pearson correlation of 0.8853 between the feature-based distance and mean SSIM across z-stacks. Scale bar = 15 μm in **a**.

Figure R2. Unmixing results of the public human cell dataset from Allen Institute for Cell Science: proteins paired with cell membrane. **a-h**, Representative image showing gray scale network input (synthetic images) and corresponding channel-wise merged network output (unmixed images) using a green-and-magenta color scheme. For each pair, the target protein is shown in green and the cell membrane in magenta. Protein pairs shown: **a**, (alpha-tubulin, cell membrane); **b**, (desmoplakin, cell membrane); **c**, (lamin B1, cell membrane); **d**, (myosin IIB, cell membrane); **e**, (Sec61 β , cell membrane); **f**, (ST6GAL1, cell membrane); **g**, (TOMM20, cell membrane); **h**, (ZO-1, cell membrane). Scale bar = 10 μm in **a-h**.

Figure R3. Unmixing results of the public human cell dataset from Allen Institute for Cell Science: proteins paired with DNA. a-h, Representative image showing gray scale network input (synthetic images) and corresponding channel-wise merged network output (unmixed images) using a green-and-magenta color scheme. For each pair, the target protein is shown in green and the DNA in magenta. Protein pairs shown: **a,** (alpha-tubulin, DNA); **b,** (desmoplakin, DNA); **c,** (lamin B1, DNA); **d,** (myosin IIB, DNA); **e,** (Sec61 β , DNA); **f,** (ST6GAL1, DNA); **g,** (TOMM20, DNA); **h,** (ZO-1, DNA). Scale bar = 10 μ m in **a-h.**

4. A similar question could be asked for membrane based cell type markers, which may not normally overlap in the same cell, but might have partial overlaps in the tissue when these cells are next to each other (and the abundance of these

incidents would vary with the region of the tissue). An example case for this would be immune cells in a tonsil sample. How would the approach deal with these kinds of situations?

(This comment is also closely related to our response to major comment 3.)

During the protein pairing process, the feature extraction network is designed to identify protein pairs with distinct spatial expression patterns. When target cells exhibit different spatial characteristics, for example different cell types with distinct morphologies, and the image resolution is sufficient to capture these distinctions, their membrane markers can be naturally distinguished despite physical proximity. In cases where partial overlaps occur, the protein separation network leverages three-dimensional protein profiles, providing enhanced spatial context for accurate unmixing. Importantly, membrane markers targeting cells with similar spatial patterns are unlikely to be paired in the first place, as our feature-based distance metric prioritizes proteins with distinguishable patterns for optimal unmixing performance.

5. If the reference images for training of the feature extraction network were coming from the same sample and field of view, one could very easily expect a good predictability from the distance matrix. However, when images are from different cells and regions (which seems to be the case here based on Fig. 2 and Supplementary Fig. S1), it is unclear to me how randomness of the region selection would affect the extracted features, and how big the dataset needs to be to compensate for heterogeneity in signal distribution especially in tissue samples like the brain where cell type abundances and noise can vary across the tissue, and especially when there is no cell segmentation etc.

As the reviewer pointed out, if both training and test datasets for the feature extraction network were obtained from the same sample and field of view, the predictability of the distance matrix could be misleadingly high due to the lack of variability.

To address this concern, we compared the feature-based distance matrices from separate training and test datasets from different samples (Figure R4). The analysis shows that the two matrices exhibit highly similar patterns, with a Pearson correlation coefficient of 0.9984. This indicates that, while the randomness in region selection could potentially affect feature extraction, this should not be a major concern in practice. Additionally, we would like to emphasize that the feature-based distances are used only to avoid the "worst pairs" (defined as the pairs with the lowest feature-based distances), which are difficult to unmix. This approach makes the method very robust against the heterogeneity in signal distribution.

This is now added as Supplementary Figure 10 in the supplementary materials.

Figure R4. Comparison of feature-based distance matrices between the separate training and test datasets. a-b, The

feature-based distance matrices calculated from two separate datasets are displayed as heatmaps with a red-yellow-blue color scheme, where blue indicates higher and red indicates lower feature-based distances. **a**, Training dataset; **b**, Test dataset. The matrices show pairwise feature-based distances between various protein markers: calbindin2, calnexin, doublecortin, GFAP, lamin B1, MAP2, NeuN, nucleolin, PV, and S100B. The strong agreement between training and test matrices with a Pearson correlation coefficient of 0.9984 demonstrates that feature extraction remains stable across different samples.

6. If utilizing singleplex images from different samples (not obtained from the same tissue slice or cells) is sufficient to calculate reliable pairwise distance, can authors utilize an existing data set (such as human protein atlas), where reference images are available? If not, what are the limitations for this?

(This comment is also closely related to our response to major comment 3.)

We thank the reviewer for this great comment which allowed us to explore a simple yet powerful way to verify the generality of our method. By testing our method on the public human cell dataset from Allen Institute for Cell Science, we confirmed that our approaches work well on a dataset that is acquired from different species (human) with different imaging procedures.

As shown in Figure R1d-e, our analysis demonstrated strong correlations between feature-based distances and unmixing performance (Pearson correlations of 0.8225 and 0.8853 for cell membrane and DNA pairs, respectively), indicating that our approach can effectively leverage data from existing sources to predict unmixing performance. We note that we are not looking for the correlation values that are close to one; these are for checking whether the feature distance is a good enough precursor of the unmixing performance.

This is now added as Figure 5 in the manuscript.

7. Suppl Fig. 1: I'd be curious to see patch examples for the non-overlapping parts of the same clusters (for example nucleolin/NeuN) - how come there are clearly separated parts of these clusters although images suggest that they would have very similar staining patterns overall.

We have added visualization of image patches for the non-overlapping parts of nearby clusters in Figure R5, including two examples: the doublecortin-GFAP (Figure R5b-d) and the nucleolin-NeuN (Figure R5e-i). Both pairs are located closely in the t-SNE plot due to their similar spatial expression patterns. Image patches corresponding to the overlapping regions of the t-SNE plot are likely indistinguishable, whereas patches from the non-overlapping regions show more distinguishable patterns.

Specifically, for doublecortin and GFAP, image patches corresponding to regions closer to the overlap display a fibrous pattern (Figure R5c). However, moving away from the overlapping region, doublecortin is observed to be expressed in the soma (Figure R5b), and GFAP shows a denser expression pattern compared to the overlapping region (Figure R5d).

For nucleolin and NeuN, image patches from the overlapping region of the two clusters show lower cell density, where the information available to distinguish two proteins is limited (Figure R5g). Conversely, moving further away from the overlapping region, image patches exhibit higher cell density, making it easier to clearly observe the difference in the spatial expression pattern of two proteins (Figure R5e,f,h,i).

This is now added as Supplementary Figure 2 in the supplementary materials.

Figure R5. Visualization of protein images that are closely located in the feature domain—non-overlapping region to overlapping region. **a**, The t-SNE plot of extracted feature vectors from each protein image patch and boxed regions showing protein pairs that are closely located in the feature domain due to their similar spatial expression patterns. **b-i**, Image patches corresponding to the boxed region in **a**. **b-d**, Doublecortin and GFAP. **b**, Doublecortin in the non-overlapping region; **c**, Doublecortin and GFAP in the overlapping region; **d**, GFAP in the non-overlapping region. **e-i**, Nucleolin and NeuN. **e-f**, Nucleolin in the non-overlapping region; **g**, Nucleolin and NeuN in the overlapping region; **h-i**, NeuN in the non-overlapping region. Scale bar = 10 μm in **b-i**.

8. The above question also points to a critical weakness of the method. It would very hard to unmix proteins that are spatial overlapping in the same cell when a single fluorophore is used. The feature vector may find some high distance between them if the weight of the features is not adjusted to avoid scenario because some features may just not be accessible for unmixing when the same fluorophores are used. Similarly other features like absolute signal intensity/ target abundance etc. may not be properly represented in the feature matrix because training images are not taken under comparable conditions (same fluorophore, same imaging settings etc.) - (I presume this is the case because no information is provided about how these stainings and imaging was done, which has to be included in the Methods. Under these

circumstances it is really not clear to me how good the feature-based distance can actually accurately capture the pairs for best unmixing success.

We would like to emphasize that both the feature extraction network and the protein separation networks are trained to "distinguish" different proteins based on their spatial expression pattern. This task alignment ensures that a pair of proteins with a large feature distance is easily unmixable by the protein separation network. We note that this training and pairing processes is entirely automatic and does not require careful adjustments of parameters. In addition, since the brightness can depend largely on the staining condition, we deliberately introduced "brightness augmentation" during the training procedure as described in the last paragraph of the first result section (SEPARATE: pairing and unmixing proteins with distinctive spatial expression patterns).

In addition, our method maintains effective unmixing capabilities even when proteins are spatially superimposed, provided they exhibit distinguishable three-dimensional expression profiles (Figure R12). The primary challenge in unmixing spatially overlapping proteins arises specifically when proteins show broad, diffuse characteristics in their overlapping regions; these regions represent background patterns rather than distinctive spatial features. While our framework primarily focuses on proteins with distinctive spatial expression patterns, in cases where proteins show diffuse overlapping characteristics, autofluorescence removal² or background subtraction can be performed as preprocessing steps to enhance unmixing.

Furthermore, we have implemented methodological approaches ensuring that feature extraction and feature-based distance calculations are primarily influenced by spatial expression patterns rather than experimental conditions. A key component is the self-supervised denoising technique, which minimizes the dependency on varying noise levels across different images. To evaluate the importance of this preprocessing step, we conducted an experimental comparison across three conditions: raw noisy data, denoised data, and denoised data with controlled noise addition. Starting with raw noisy data (Figure R6a with a calbindin 2 as a representative example), we applied self-supervised denoising method and then introduced Gaussian noise with standard deviations of k times the standard deviation σ_i of denoised image, where k ranged from 0.1 to 1.9 in steps of 0.2. We trained individual feature extraction networks for each dataset condition. The pairwise feature-based distance matrix showing the relationships between different protein markers in raw noisy image is displayed in Figure R6b. The cross-correlation between the feature-based distance matrices obtained from each experiment at raw noisy data, denoised data, and different noise levels is shown in Figure R6c. The cross-correlation reveals that raw noisy data shows relatively lower correlation, below 0.8, highlighting the potential inconsistency of noise levels between different protein images, which can lead the feature extraction network to distinguish protein images based on different noise level rather than the difference in spatial expression patterns. In contrast, the consistently high correlations, around 0.9, maintained between denoised and controlled noise conditions demonstrate that self-supervised denoising preprocessing step effectively addresses this challenge by normalizing noise characteristics across all proteins. Representative images of calbindin 2, showing the noise level of denoised and incremental Gaussian noise added images, and their corresponding feature-based distance matrices are displayed in Figure R6d.

This is now added as Supplementary Figure 13 in the supplementary materials.

Figure R6. Effects of noise levels on feature-based distance matrices. **a**, Raw noisy image showing the spatial expression pattern of calbindin 2 as a representative example among the 10 protein markers. **b**, The pairwise feature-based distance matrix calculated from the feature extraction network for raw noisy images of 10 protein markers (calbindin 2, calnexin, doublecortin, GFAP, lamin B1, MAP2, NeuN, nucleolin, PV, S100B), displayed as a heatmap with a red-yellow-blue color scheme, where blue indicates higher and red indicates lower feature-based distances between protein pairs. **c**, Cross-correlation analysis between feature-based distance matrices at different noise levels, showing the relationship between raw noisy, denoised, and noise-added conditions. Yellow indicates a higher correlation, while pink indicates a lower correlation. **d**, Representative images of Calbindin 2 showing the noise level of denoised and incremental Gaussian noise added images ($k \times \sigma_i$, $k = 0.3$ to 1.9, where σ_i represents the standard deviation of the denoised image). The corresponding pairwise feature-based distance matrices from individually trained feature extraction networks with each condition are shown below. Scale bar = 30 μm in **a** and **d**.

9. Could the authors please explain the t-SNE in Fig. 2 better? Is each dot one patch of one image/FOV? In the Reporting Summary document, they mention three datasets were used for training and 1 dataset was used for feature extraction. What exactly is one dataset and how big they need to be for reproducible feature extraction? If it was a single dataset that was utilized for analysis, how different would the data in Fig. 2b-c look for an independent repetition (same markers different sample/ imaging region)?

We have revised the manuscript to provide a more detailed explanation of the t-SNE plot in Figure 2. Each dot in the t-SNE plot represents a feature vector corresponding to a single image patch used as input for the feature extraction network. Each image patch is 256 \times 256 pixels, corresponding to an area of 51.2 $\mu\text{m} \times$ 51.2 μm . Representative image patches are shown in Supplementary Figure 1.

Additionally, a 'dataset' refers to a single z-stack image from one imaging session, consisting of multiple z slices. Each z-stack image has dimensions of 1024 (x) × 1024 (y) × 26 (z) pixels, corresponding to a volume of 204.8 μm × 204.8 μm × 25 μm. We used three z-stack images for training the feature extraction network for each protein and one for analysis (Figure R7a). This detailed information has been included in the Methods section and the Reporting Summary.

To assess reproducibility, we conducted a cross-validation analysis by rotating the roles of our four datasets between training the feature extraction network and calculating feature-based distances to identify optimal grouping (Figure R7b). Rather than directly comparing optimal groupings, we analyzed the relationships between these distance matrices across experiments. This is because the relative relationships and trends between feature-based distances are more informative than their absolute values, as the feature extraction network incorporates randomness during the training process to ensure robustness. These randomization strategies help prevent overfitting and ensure the network learns generalizable features rather than memorizing specific patch orientations or positions. While these random elements may cause slight variations in the absolute feature-based distances across different training iterations, the relative relationships between protein pairs remain consistently preserved.

The pairwise feature-based distance matrices from each experiment are shown in Figure R7c. Figure R7d presents a correlation map across experiments and the corresponding box-and-whisker plot of correlation values ($n = 6$). The correlation coefficients above 0.8 demonstrate that our feature extraction approach reliably captures the underlying protein relationships regardless of the training and optimal pair identification dataset combinations.

This is now added as Supplementary Figure 11 in the supplementary materials.

Figure R7. Cross-validation on the reproducibility of feature extraction across different dataset combinations. a, Schematic representation of the z-stack datasets. Each protein (1 to N) has four z-stack images, where each z-stack consists of multiple image slices. Each z-stack of protein images is shown in different colors to indicate separate imaging sessions. For each protein, three z-stacks were used to train the feature extraction network, while one was used to identify optimal pairs. **b**, Overview of the cross-validation scheme, showing how four z-stack images were alternately used for network training (colored text) and optimal pair identification (black text) across different experiments. **c**, The pairwise feature-based distance matrices calculated from each experiment, are displayed as heatmaps with a red-yellow-blue color scheme, where blue indicates higher and red indicates lower feature-based distances between protein pairs. The consistency in relative patterns across matrices

demonstrates the robustness of our approach across different experiments. **d**, Cross-correlation analysis between feature-based distance matrices, with the heatmap showing the correlation between experiments (left) and their distribution summarized in a box-and-whisker plot (right, $n = 6$). Yellow indicates a higher correlation, while pink indicates a lower correlation. The boxes show the interquartile range (IQR) with the median, while whiskers extend to 1.5 times the IQR. Individual points represent outliers. The correlation values above 0.8 confirm that the relative relationships between protein pairs are preserved regardless of the training and optimal pair identification dataset combinations.

10. Similar to the point above, what are n (sample sizes) and the error bars for different graphs? Is almost all of the main data that went into the manuscript generated by imaging just one field of view for each target, and z positions are used for getting the sample number to have error bars? This is not clear for Fig. 3e-f. Unless the z -step sizes are really big, distributions from one position to the next might be quite similar at least for some targets, which would be an argument to not treat them as different samples for statistics.

We appreciate the reviewer's careful attention to the statistical details of our analysis. To address these points, we have thoroughly revised our presentation of the data and statistical analysis. In the revised manuscript, we have added detailed information about sample sizes and specifics of the box-and-whisker plots in all relevant figures, including clearly labeling n values and what they represent for each dataset.

Regarding our data collection methodology for Figure 3, which demonstrates experimental validation of protein pairing using the feature extraction network for fluorescent signal unmixing performance, we want to clarify that we collected four z -stack images for each pair, where three stacks were allocated for training while one stack was reserved for testing, and we acknowledge the reviewer's concern about treating each z -position as an independent sample for statistical analysis.

We have addressed this by dividing the entire image into 16 non-overlapping patches, treating each patch as an independent sample for the graph representation. For the analysis of the public human cell dataset from Allen Institute for Cell Science presented in Figure R1, which contained a larger number of image stacks, we were able to utilize each image stack as an independent sample, providing a more robust statistical representation.

11. Even for markers that are spatially distant, the unmixing performance is not great. For example in the network output for PV one could clearly see the remnants of Lamin B1 which is not visible in the ground truths in fig. 3. Based on this, how is Fig. 4 result should be evaluated? At the lower right corner of the Fig.4f I could clearly see red nuclei with bright nucleoli - these are assigned to Doublecortin, however doublecortin is found in the cytoplasm, and the unmixing probably failed by wrong assigning nucleolin to doublecortin. Interestingly, this is the pair, where SSIM was highest.

We appreciate the reviewer's careful observation regarding the unmixing results. First, regarding the unmixing performance between lamin B1 and PV, we acknowledge that the relative brightness of proteins within a single channel impacts unmixing success. To address this challenge during network training, we deliberately varied the mixing coefficients in synthetic image generation. Specifically, we set a minimum coefficient of 0.01 and constrained the sum of two coefficients to range between 0.8 and 1.2, enabling the network to handle brightness ratios up to approximately 100-fold between proteins. As demonstrated in Figure R8 with lamin B1 and PV pair, our network maintains robust performance ($SSIM > 0.7$) across varying signal ratios, although the unmixing performance slightly decreases with larger differences in relative brightness between proteins.

Regarding the (doublecortin, nucleolin) pair observed in the lower right corner of Figure 4f, the appearance in the merged visualization reflects the distinct spatial distributions and varying expression levels of these proteins. While the merged view might suggest potential misallocation due to intensity differences, a detailed examination of the individual channels confirms proper protein localization. To better illustrate this point, we provide unmixing results with single-channel visualizations in Figure R9.

Figure R8 is now added as Supplementary Figure 20 in the supplementary materials.

Figure R8. Comparison of unmixing performance across different relative brightness of protein signals. a, Representative images showing unmixing results for varying signal ratios between two proteins, lamin B1 and PV, for three different mixing coefficients ($0.3I_A + 0.7I_B$, $0.5I_A + 0.5I_B$, $0.7I_A + 0.3I_B$, where I_A represents the image of lamin B1 and I_B represents the image of PV). The unmixing results are overlaid with ground-truth images using a red-and-cyan color scheme to enable intuitive interpretation: white indicates correctly unmixed signals, red shows false positive assignments, and cyan represents false negative assignments. Top row shows the comparison between unmixed and ground-truth images for lamin B1, and bottom row shows the comparison for PV. **b,** Quantitative assessment of unmixing performance using SSIM for varying mixing coefficient combination. Box plots represent the SSIM of non-overlapping patches ($n = 16$) for each signal mixing coefficient. The boxes show the interquartile range (IQR) with the median, while whiskers extend to 1.5 times the IQR. While the performance slightly decreases with larger differences in relative brightness between proteins, the SSIM values consistently remain above 0.7, indicating robust unmixing performance across varying signal ratios. Scale bar = 30 μm in **a**.

Figure R9. Visualization of unmixed doublecortin and nucleolin signals. **a**, Single-channel and channel-wise merge unmixing results of (doublecortin, nucleolin) pair. The top row shows the full field of view unmixing images of doublecortin and nucleolin in single-channel and channel-wise merge using a green-and-magenta color scheme (doublecortin in green, nucleolin in magenta). The bottom row displays magnified views of the region highlighted by yellow boxes. Scale bar = 20 μm in the top row and 5 μm in the bottom row of **a**.

12. The SSIM data is the main benchmark in the paper, however it could be seen that even for the cases of pairs with low distances SSIM values are quite high (over >0.9), suggesting this may not be the best measure. It is obvious that SSIM would not be very sensitive for the actual unmixing performance when highly overlapping distributions (such as nucleolin/NeuN) are being evaluated, as the NeuN similarity to nucleolin groundtruth and vice versa would already be very high. Should this at least not be also plotted or used for normalization.

We appreciate the thoughtful comments regarding our use of SSIM as a benchmark metric. In line with major comment 10, we have revised our analysis methodology by treating non-overlapping patches as independent samples rather than considering each z-position independently. This modification in the sampling approach has resulted in slightly different SSIM values as shown in Figure R10a compared to our previous analysis. This analysis demonstrates the effectiveness of feature-based distance as a predictor of unmixing performance, with a Pearson correlation coefficient of 0.7566 between the distance measures and SSIM values.

Regarding the specific case of nucleolin and NeuN pair, we acknowledge the value of cross-similarity between two target images in providing additional insights. We have included cross-similarity plots between these markers in Figure R10b, and the analysis showed a strong negative correlation between cross-similarity and feature-based distance with a Pearson correlation coefficient of -0.7975 . To visually emphasize this negative correlation, we inverted the feature-based distance axis and displayed it in red. We then normalized SSIM values using the baseline cross-similarities as suggested (Figure R10c), employing the formula $\frac{\text{SSIM}}{1 + \text{cross SSIM}}$ to avoid large deviation from near-zero denominators. This normalized analysis strengthened the relationship with feature-based distance, achieving a Pearson correlation coefficient of 0.8914. These findings indicate that the normalization approach effectively quantifies the relative unmixing performance across protein pairs.

To provide complementary evaluation metrics, we extended our analysis using the Pearson correlation coefficient (PCC) as shown in Figure R10d-f. Similar to SSIM, the PCC analysis revealed a positive correlation with the feature-based distance with the correlation coefficient of 0.7609. Following our cross-similarity approach, we analyzed cross-PCC relationships, which demonstrated a negative correlation with the feature-based distance, yielding the correlation coefficient of -0.6601 . When normalized using the formula $\frac{\text{PCC}}{1 + \text{cross PCC}}$ to avoid large deviations from near-zero denominators, the PCC values showed an improved correlation with feature-based distance with the correlation coefficient of 0.7733.

This is now added as Supplementary Figure 19 in the supplementary materials.

Figure R10. Relationship between feature-based distances and unmixing performance metrics. a-f, Correlation analysis between feature-based distance and unmixing performance using SSIM and PCC as metrics. Each box plot represents the SSIM or PCC values of non-overlapping patches ($n = 16$) for the corresponding protein pair, with the feature-based distances shown as a purple dotted line. The boxes show the interquartile range (IQR) with the median, while whiskers extend to 1.5 times the IQR. Individual points represent outliers. **a**, Correlation analysis between feature-based distances and SSIM values for unmixed protein pairs, demonstrating the predictive power of feature-based distances for unmixing performance with the Pearson correlation coefficient of 0.7566; **b**, Cross-similarity analysis showing the negative correlation between cross-SSIM and feature-based distances with the Pearson correlation coefficient of -0.7975 , where the feature-based distance axis is inverted and displayed in red; **c**, Normalized SSIM analysis (normalized by $\frac{\text{SSIM}}{1 + \text{cross SSIM}}$ to avoid large deviations from near-zero denominators) revealing enhanced correlation with feature-based distances, yielding a Pearson correlation coefficient of

0.8914, validating the effectiveness of our feature-based approach in predicting unmixing performance; **d**, PCC analysis exhibiting comparable trends to SSIM, showing positive correlation with feature-based distances with the Pearson correlation coefficient of 0.7609; **e**, Cross-PCC analysis exhibiting negative correlation between feature-based distances, showing a Pearson correlation coefficient of -0.6601 , where the feature-based distance axis is inverted and displayed in red; **f**, Normalized PCC analysis (normalized by $\frac{\text{PCC}}{1 + \text{cross PCC}}$ to avoid large deviations from near-zero denominators) showing improved correlation with feature-based distances, obtaining the Pearson correlation coefficient of 0.7733.

13. How do different factors, such as resolution, autofluorescence and noise of the reference images affect the assignment of pairs? Authors can at least process the same data (for example by binning) to see how the performance is getting affected. Authors mention in the methods that they apply self-supervised denoising to the images before using them for training, how necessary is this? And is it critical for getting good performance?

We appreciate the reviewer's important question regarding the robustness of our method to various imaging factors. To address these concerns, we conducted additional experiments examining how resolution and noise levels affect the identification of optimal grouping.

As addressed in our response to major comment 9, we evaluated the identification of optimal grouping by analyzing the correlation between pairwise feature-based distance matrices across different experimental conditions. We chose this metric because while small variations in distance values might lead to different specific groupings, the overall relative relationships between distances remain stable. This correlation-based assessment provides a more reliable measure of our method's consistency than group-level assessment.

We investigated the effects of spatial resolution by applying progressive binning factors (2, 4, 8, 16, 32, and 64) to our dataset, effectively simulating varying imaging resolutions. Using calbindin 2 as a representative example, Figure 11a shows the original resolution image, while Figure R11d demonstrates how increasing binning factors progressively reduce spatial detail. The pairwise feature-based distance matrix corresponding to the original resolution images is displayed in Figure R11b.

The cross-correlation between feature-based distance matrices at different resolutions (Figure R11c) demonstrates remarkable stability of our method. Binning factors up to 16 maintain high correlation values, above 0.85 with the original resolution, indicating that the essential features of protein expression patterns are preserved even at reduced resolution. However, at binning factors of 32 and 64, where cellular structures become severely pixelated, the correlation values drop substantially, to under 0.7, indicating a threshold beyond which spatial information becomes insufficient for reliable feature extraction.

These findings demonstrate that our feature extraction method performs reliably across a practical range of imaging resolutions, maintaining consistent relationships between the spatial expression patterns of proteins as long as basic cellular morphology remains discernible. This robustness to resolution variation suggests that our method could be applicable across different imaging platforms and experimental conditions without requiring stringent standardization of spatial resolution.

Similarly, to assess the impact of noise and the self-supervised denoising, we introduced controlled levels of Gaussian noise to our dataset (Figure R6). As detailed in our response to major comment 8, this analysis was crucial in demonstrating how the self-supervised denoising preprocessing step helps ensure that feature extraction is driven by spatial expression patterns rather than experimental noise variations.

This is now added as Supplementary Figure 12 in the supplementary materials.

Figure R11. Effects of spatial resolution on feature-based distance matrices. **a**, Original resolution image showing the spatial expression pattern of calbindin 2 as a representative example among the 10 protein markers. **b**, The pairwise feature-based distance matrix calculated from the feature extraction network for original resolution images of 10 protein markers (calbindin 2, calnexin, doublecortin, GFAP, lamin B1, MAP2, NeuN, nucleolin, PV, S100B), displayed as a heatmap with a red-yellow-blue color scheme, where blue indicates higher and red indicates lower feature-based distances between protein pairs. **c**, Cross-correlation analysis between feature-based distance matrices at different binning factors (2, 4, 8, 16, 32, and 64), showing the relationship between different spatial resolutions. Yellow indicates a higher correlation, while pink indicates a lower correlation. **d**, Representative images of calbindin 2 showing the spatial resolution with different binning factors. The corresponding pairwise feature-based distance matrices from individually trained feature extraction networks with each condition are shown below. Scale bar = 30 μm in **a** and **d**.

14. When two different targets are combined into one imaging channel, they need to have similar brightness so that one acquisition setting could give sufficient but not saturating signal for both targets. This would be a critical consideration for how to make pairs. This is not mentioned or discussed at all, and definitely needs to be addressed. And if the networks were somehow accounting for this, it would be actually very useful.

We agree with the reviewer that the relative brightness between two targets in a single imaging channel is a critical consideration for optimal imaging and unmixing. Our investigation of this issue, detailed in response to major comment 11, led us to incorporate brightness considerations into our network training process. The protein separation network is trained to handle brightness ratios up to 100-fold between two proteins, although optimal unmixing performance is achieved when the brightness levels of the two proteins are comparable. We have now added this important consideration to our protein pairing strategy discussion in the Methods section.

15. Line 171-175: Authors discuss the requirement of 3D images for training. What thickness is relevant (not mentioned in the manuscript). Would 2D projections of 3D stacks not be sufficient for training? Projections usually capture the overall signal morphology (such as fibrous versus spotty). Related to this and imaging process, many relevant technical details are omitted. According to the Methods, sections are 150 μm thick, but based on the images it seems like maybe only the upper layers were imaged. It is unclear what the actual thickness is and what kind of z-step size is used.

We appreciate the reviewer's insightful questions regarding the z-stack requirements and imaging specifications. To address the importance of 3D information, we conducted experiments using a representative protein pair (calbindin 2, calnexin), comparing 3D-trained models with varying z-stack depths to 2D projection trained models.

We first evaluated how the number of z-slices affects the performance of 3D-trained models. The unmixing results with different z-stack depths (16, 8, and 2 z-slices) are presented in Figure R12a, where we visualize a single representative z-position to demonstrate how the quality of unmixing is affected by the number of z-slices used during training. For detailed examination, magnified views of regions highlighted by yellow boxes are shown alongside each condition. The corresponding unmixing performances are quantified using box plots in Figure R12b. The quantitative analysis shows that performance metrics quantified with SSIM decrease as the z-depth is reduced, with the highest SSIM values observed for 16 z-slices and a notable decline when using only 2 z-slices.

Building upon these findings, we extended our analysis to compare 3D-trained models with a 2D projection trained model (Figure R13). While 2D projections might seem adequate for capturing overall signal morphology, these projections inherently compress spatial information, causing signal overlap from different z-planes that compromises accurate protein expression patterns. To demonstrate this limitation, we present the same 3D unmixing results as in Figure R12, but visualized as maximum intensity projections (MIP) to enable direct comparison with results from 2D projection trained model. This comparison is possible because our 3D-trained model can generate results for the entire z-stack, which can then be projected into 2D, whereas models trained on 2D projections can only output 2D results. As evident in the magnified regions highlighted by yellow boxes, the model trained on 2D projections shows significant loss of critical spatial information and protein expression patterns compared to the projected results from 3D-trained models, particularly in areas where multiple proteins are co-localized.

Regarding the technical specifications, while our tissue sections are 150 μm thick, as described in the Methods section, we utilized approximately 25 μm depth for imaging as the deeper area showed limited SNR due to the antibody penetration issue. We employed a z-step size of either 0.5 or 1 μm .

This is now added as Supplementary Figures 16 and 17 in the supplementary materials and the Method section in the manuscript.

Figure R12. Comparison of unmixing performance of 3D-trained models across varying z-stack depths. Analysis of z-stack depth effects on protein unmixing performance, demonstrated with (calbindin 2, calnexin) pair as representative protein pair. **a**, Ground truth and unmixing results at a single z-position shown in channel-wise merge using a green-and-magenta color scheme (calbindin 2 in green, calnexin in magenta) for 3D-trained models across varying z-stack depths (16, 8, and 2 z-slices). Magnified views of regions highlighted by yellow boxes are shown on the right of each condition. **b**, Box-and-whisker plots of unmixing performance quantified by SSIM demonstrating decreased performance with reduced z-depth ($n = 16$, non-overlapping 3-dimensional z-stack patches). The boxes show the interquartile range (IQR) with the median, while whiskers extend to 1.5 times the IQR. Scale bar = 30 μm in full-field images and 5 μm in magnified views in **a**.

Figure R13. Comparison of 3D-trained models with 2D projection trained model. Demonstration of 3D-trained model advantages using (calbindin 2, calnexin) pair as a representative protein pair. **a**, Maximum intensity projections (MIPs) of ground truth and unmixing results shown in channel-wise merge using a green-and-magenta color scheme (calbindin 2 in green,

calnexin in magenta) comparing 3D-trained models with varying z-stack depths (16, 8, and 2 z-slices) to 2D projection trained model. Magnified views of regions highlighted by yellow boxes are shown below each condition. Scale bar = 30 μm in full-field images and 5 μm in magnified views in **a**.

16. The authors are going with a relatively small set of markers right now. How does the method scale when more protein markers are included?

In SEPARATE, we employ feature-based distances between protein markers to predict unmixing performance, with our algorithm optimizing the worst-case scenario by maximizing the minimum distance between paired markers in grouping. To assess scalability with larger marker sets, we conducted Monte Carlo experiments with progressively larger marker sets, increasing k from 4 to 60. Our simulation framework randomly generated k points within a unit circle, where each point represents a protein marker in the normalized feature space. For each configuration, we identified the optimal grouping that maximizes the minimum distance between pairs. Figure R14a illustrates the example of Monte Carlo experiment with 6 points. We repeated this process for 2,000 independent trials to ensure statistical robustness.

The results demonstrate that stability of SEPARATE in identifying optimal grouping improves with larger marker sets. As shown in Figure R14b, as k increased, the median of the maximized minimum distance shows an upward trend while the variance decreases, indicating more stable and reliable identification of optimal groupings with larger marker sets. These findings, now included as Supplementary Figure 18, indicate that SEPARATE not only scales well but potentially performs better with larger marker panels by achieving more stable and reliable groupings.

Figure R14. Scalability analysis of SEPARATE with an increasing number of protein markers through Monte Carlo experiments. **a**, Illustration of the Monte Carlo experiment setup. Random points ($k = 6$) are generated within a unit circle, and optimal grouping that maximizes the minimum distance between paired points is identified. The paired points are connected by dashed lines, with distances d_1 , d_2 , and d_3 . **b**, Box-and-whisker plots of maximized minimum distance for 2,000 independent trials with an increased number of points ($k = 4$ to 60). The boxes show the interquartile range (IQR) with the median, while whiskers extend to 1.5 times the IQR. Individual points represent outliers. The results show increasing median values and decreasing variance as k increases, demonstrating that SEPARATE achieves more stable and reliable groupings with larger marker sets.

17. Methods section: For prelabeling of antibodies with Fabs, the Fab-antibody molar ratio is critical. In the methods section only a volumetric ratio is given, which is not informative. When the antibody-Fab binding is not saturated there is a big risk of crosstalk, which would affect the results for this kind of a study (especially here all the antibodies are from

one source species). Additionally, antibodies and clones (vendors, source species, dilutions) need to be also reported in the main text.

We apologize for the omission of these important details. Here, we addressed each point raised by the reviewers individually.

1. Fab-antibody molar ratio

The pre-labeling of primary antibodies with Fab fragments was first proposed by Brown et al. (2004, *J. Histochem. Cytochem.* 52(12):1563–1576) and subsequently validated in our previous study (Seo et al., 2022, *Nat. Commun.* 13:2475). We adhered to the protocols outlined in these studies with minor modifications. During the labeling process, we fixed the volume ratio of 1× PBS, a solution containing a fluorophore-conjugated Fab fragment antibody, and a solution of primary antibody to 10:2:1, even though the concentrations of primary antibodies provided by commercial vendors vary. This corresponds to a molar ratio of Fab fragment antibody to primary antibody of 6:1 for a primary antibody concentration of 1 mg/mL.

The revised Methods section now specifies:

“Preformed antibody complexes were prepared following the original primary antibody-Fab complex formation protocol³. Specifically, 1× PBS, a solution containing a fluorophore-conjugated Fab fragment antibody, and a solution of primary antibody were mixed in volumes of 10 μL, 2 μL, and 1 μL, respectively. This volume ratio corresponds to a molar ratio of Fab fragment antibody to primary antibody of 6:1 for a primary antibody concentration of 1 mg/mL. The mixture was then incubated for 10 minutes at RT in the dark. Then, 187 μL of blocking buffer (5% normal rabbit serum, 0.1% Triton X-100, 1× PBS) was added to the solution and incubated for 10 minutes at RT in the dark with gentle shaking. For staining, additional blocking buffer was added to make a final primary antibody dilution ratio of 1:1000. For staining, brain slices were incubated in the diluted antibody solution at 4°C for overnight in the dark.”

2. Crosstalk mitigation

As highlighted by Brown et al. (2004), incomplete saturation during Fab-antibody binding can result in crosstalk, where secondary antibodies interact with unbound primary antibodies. This phenomenon causes nonspecific staining, leading to overlapping signals and obscuring specific antigen detection. Such interference could compromise the accuracy of our results, particularly in multiplex imaging experiments. To address this, Fc fragment-specific Fab fragments were used, as they have fewer binding sites on primary antibodies and can be easily saturated, ensuring clear signal unmixing while minimizing potential crosstalk.

Given that all primary antibodies used in this study originated from the same species, we implemented additional measures to reduce crosstalk. First, as described above, the Fab-antibody binding process was optimized to achieve saturation. Second, we used a blocking buffer containing 5% normal rabbit serum, 0.1% Triton X-100, and 1× PBS during the staining process to minimize nonspecific binding. After forming the preformed Fab-antibody complexes, the complexes were diluted in the blocking buffer and applied to the samples at 4°C overnight in the dark. These procedures align with the strategies initially developed by Brown et al. (2004) and validated by Seo et al. (2022), demonstrating effective multiplex protein imaging with minimal signal interference.

3. Detailed antibody and clone information

Detailed information on all antibodies used in the study, including vendors, source species, clone details, and dilution ratios, has been added to Supplementary Table 2. A reference to this table has been included in the main text for improved transparency and reproducibility.

Minor Comments

- Intro Line 44: Authors state: "However, as staining typically takes more than 12 hours and can extend to several days for thick specimens²³, performing multi-round staining and imaging could significantly prolong the entire process." Imaging cycles could be much shorter (down to 15-30 min with thin sections and a couple of hours for thicker samples) for methods that do not require antibody relabeling, such as DNA oligo barcoding.

We appreciate the reviewers' comments and agree that staining times can be shorter for certain conditions. For example, 15–30 minutes is sufficient for staining formalin-fixed paraffin-embedded (FFPE) specimens with a thickness of less than 5 μm . However, for 3D imaging of thicker specimens, longer staining times are necessary. For example, recently published work on 3D multiplexed imaging of 35- μm -thick FFPE specimens reported staining durations of 8 to 10 hours (Yapp C, et al. Multiplexed 3D Analysis of Cell Plasticity and Immune Niches in Melanoma. *bioRxiv* (2023). doi:10.1101/2023.11.10.566670). For even thicker specimens, such as those 150 μm in thickness, staining may need to be extended to overnight or multiple days. Such prolonged staining times not only lengthen the overall process but also compromise specimen integrity and antigenicity. We believe that our technique, which reduces the number of cycles by half, provides a significant advantage in addressing these challenges. Accordingly, we have revised the introduction as follows:

“In two-dimensional imaging of thin tissue slices, such as those with a thickness of less than 10 μm , where staining can be completed relatively quickly—even in about an hour—the repeated staining process is not a major issue. However, specimens with a thickness beyond 10 μm , such as 35 μm , typically require much longer staining times, often extending up to 10 hours⁴. As a result, the total staining time for ten-color imaging could exceed several days. Prolonged staining time does not just extend the total process; it can also compromise specimen quality and antigen integrity^{5,6}. Spectral imaging and unmixing could provide a solution to this problem, as they allow the use of more than three fluorophores in a single staining cycle⁷. However, spectral imaging can be complex and often requires specialized equipment capable of acquiring images from various spectral ranges^{8,9}. Therefore, there is an urgent need for techniques that can reduce the total staining time for 3D multiplexed imaging.”

- What are the coefficients used for generation of synthetic images?

We used a minimum coefficient of 0.01 and ensured that the sum of the two coefficients used for image blending fell between 0.8 and 1.2 for generating synthetic images. This information has now been included in the Methods section.

- Presenting single channel microscopy data in grayscale would be more ideal (Fig. 2a; 3a-b; S1b-f).

We have updated the figures to present single-channel microscopy data in grayscale. Additionally, to maintain consistency with the t-SNE coloring scheme, we have labeled each protein name with the corresponding color instead of using the lookup table for the image.

- Use of different color combinations for every image in Fig. 3 overlays are not helping (one color combination, such as green/magenta would be sufficient)and with the chosen color schemes it is hard to see the actual signal overlap (like two blues in NeuN/nucleolin).

We have updated the coloring scheme for Figure 3c,d to improve readability and ensure color-blind friendliness, substituting it with Supplementary Figure 2, which uses a green-and-magenta color scheme.

- Plots in 3e & f are considered as main results and are too small to properly see. Fig. 3 should show input/output for all targets.

We have enlarged the plots in Figure 3e-f, and included enlarged images in Figure R15-19 all pairs of proteins.

- Developmental stage / age of sacrificed mice is missing.

Thank you for bringing this to our attention. We have revised the manuscript to include the developmental stage and age of the mice used in the study. Specifically, we clarified that C57BL/6J mice aged 4–8 weeks were utilized. The revised text now reads as follows:

“C57BL/6J mice aged 4–8 weeks were maintained in ventilated cages under standardized conditions, including a 12-hour light/dark cycle, temperatures ranging from 20–24°C, and 40–60% humidity. Before perfusion, the mice were anesthetized with isoflurane, followed by transcardial perfusion with ice-cold 4% paraformaldehyde (PFA) in 1× phosphate-buffered saline (PBS). The harvested brains were subsequently immersed in the same PFA solution at 4°C for 2 hours. Afterward, the brains were sliced into 150-µm sections using a Leica VT1000S vibratome. The slices were then stored in 0.1 M glycine and 0.01% sodium azide in 1× PBS at 4°C until further analysis.”

Additionally, we identified an error in the originally reported animal protocol number in the manuscript, which has now been corrected. The revised and approved protocol number is KA2020-48. We have ensured that this updated information is reflected accurately in the manuscript.

- Researchers testing the SEPARATE approach would benefit from parameters on how to select pairs in complex panels (i.e. what is the minimum distance that is still acceptable between targets).

Since our feature vectors are latent variables, their absolute values are less meaningful than their relative relationships. To address this, we normalized the feature space so that all feature vectors lie within a unit circle (as in Figure R14). Through our extensive experiments, we observed that when the normalized feature-based distance is greater than 1.5, achieves unmixing performance with SSIM values above 0.9.

- Would there be any limitations in the fluorophore selection?

We appreciate the reviewer’s thoughtful comments. The fluorophore selection criteria in our work are essentially identical to those considered when performing multiplexed fluorescence imaging without spectral unmixing. In this study, we selected spectrally distinct fluorophores to eliminate the possibility of signal mixing due to spectral overlap. Therefore, the primary criterion for fluorophore selection is the use of spectrally distinct fluorophores, a standard practice in many biology labs. Secondly, the fluorophores must be photostable to enable reliable 3D multiplexed imaging. The fluorophores used in this work meet these criteria and are widely employed in various multiplexed imaging studies.

- How does the computational unmixing deal with auto-fluorescence that can range from general tissue background (FFPE tissue blocks) to bright signals in select cells (blood vessels)?

Our computational unmixing algorithm does not inherently include a mechanism to handle autofluorescence. If data with significant autofluorescence are used, it would be necessary to apply an autofluorescence removal technique² prior to using SEPARATE.

- Fig4 c,d,e - protein names are not written for the respective images.

We have added protein names to the respective images for Figure 4c-e in the revised version.

- There is no mention of how the fiducials (bright, defined landmarks in a single color / dye commonly used in iterative staining protocols) could be utilized in parallel to the SEPARATE.

We would like to clarify that SEPARATE, in the current form, involves only single-round imaging, and therefore does not require fiducial markers typically used for image registration across multiple rounds of imaging. However, as the reviewer pointed out, SEPARATE can be integrated with iterative staining protocols if desired. In such cases, common nuclear stains such as DAPI in the 488 channel can serve as a fiducial marker to facilitate image registration across multiple rounds. We have incorporated this discussion about the potential integration with iterative staining protocols in the Discussion section of the manuscript.

- It would be interesting to see the performance of the computational unmixing in a subcellular panel with defined markers at higher resolution where artifacts can be more easily observed.

We have provided enlarged images in Figure R15-19 that show the unmixing results, allowing for detailed inspection of the unmixing performance and any potential artifacts.

Figure R15. Visualization of the unmixing results. a-b, Visualization of the unmixing results of two proteins. From left to right: input gray scale image, unmixed image, ground-truth image. Both unmixed image and ground-truth image represented as channel-wise merge using a green-and-magenta color scheme. **a,** Results of (calbindin 2, calnexin) pair, with calbindin 2 in green and calnexin in magenta; **b,** Results of (doublecortin, nucleolin) pair, with doublecortin in green and nucleolin in magenta. Scale bar = 30 μm in **a** and **b**.

Figure R16. Visualization of the unmixing results. a-b, Visualization of the unmixing results of two proteins. From left to right: input gray scale image, unmixed image, ground-truth image. Both unmixed image and ground-truth image represented as channel-wise merge using a green-and-magenta color scheme. **a,** Results of (GFAP, NeuN) pair, with GFAP in green and NeuN in magenta; **b,** Results of (lamin B1, PV) pair, with lamin B1 in green and PV in magenta. Scale bar = 30 µm in **a** and **b**.

Figure R17. Visualization of the unmixing results. a-b, Visualization of the unmixing results of two proteins. From left to right: input gray scale image, unmixed image, ground-truth image. Both unmixed image and ground-truth image represented as channel-wise merge using a green-and-magenta color scheme. **a,** Results of (MAP2, S100B) pair, with MAP2 in green and S100B in magenta; **b,** Results of (calbindin 2, PV) pair, with calbindin 2 in green and PV in magenta. Scale bar = 30 μm in **a** and **b**.

Figure R18. Visualization of the unmixing results. a-b, Visualization of the unmixing results of two proteins. From left to right: input gray scale image, unmixed image, ground-truth image. Both unmixed image and ground-truth image represented as channel-wise merge using a green-and-magenta color scheme. **a,** Results of (calnexin, lamin B1) pair, with calnexin in green and lamin B1 in magenta; **b,** Results of (doublecortin, MAP2) pair, with doublecortin in green and MAP2 in magenta. Scale bar = 30 μm in **a** and **b**.

Figure R19. Visualization of the unmixing results. a-b, Visualization of the unmixing results of two proteins. From left to right: input gray scale image, unmixed image, ground-truth image. Both unmixed image and ground-truth image represented as channel-wise merge using a green-and-magenta color scheme. **a,** Results of (GFAP, S100B) pair, with GFAP in green and S100B in magenta; **b,** Results of (NeuN, nucleolin) pair, with NeuN in green and nucleolin in magenta. Scale bar = 30 μm in **a** and **b**.

- Fig2c would be better to see the actual numbers for the colormap.

We have updated Figure 2c to show the actual values in the colormap of the feature-based distance matrix.

- Line 329-241: “Protein pairs with larger feature-based distances were more accurately unmixed, with a Pearson correlation of 0.8728 between the feature-based distance and the average SSIM. Scale bar = 30 μm in a-d.” Is the given PCC for the pair with highest SIM? PCC values should be given for all conditions.

For each protein pair, we calculated Pearson correlation coefficients between ground-truth and unmixing results, averaging across independent, non-overlapping z-stack patches: (calbindin 2, calnexin): 0.8406, (doublecortin, nucleolin): 0.9462, (GFAP, NeuN): 0.9377, (lamin B1, PV): 0.8863, (MAP2, S100B): 0.8157, (calbindin 2, PV): 0.8479, (calnexin, lamin B1): 0.8693, (doublecortin, MAP2): 0.8937, (GFAP, S100B): 0.9214, and (NeuN, nucleolin): 0.9076.

The previously reported Pearson correlation coefficient of 0.8728 has been updated to 0.7566 following our revised analysis methodology, which treats non-overlapping patches as independent samples rather than considering each z-position

independently (please see major comment 10). This coefficient evaluates the relationship between feature-based distances and unmixing performance quantified by SSIM across all protein pairs, suggesting that feature-based distance effectively predicts unmixing performance. We note that we are not seeking correlation values close to one, as these values simply indicate whether the feature distance serves as a reliable predictor of unmixing performance.

Reviewer #3 (Remarks to the Author):

In this manuscript Kim G. et al. describe SEPARATE, a method for multiplexed fluorescent imaging aimed at halving the time required to detect up to 10 protein targets. The authors achieve this by identifying proteins pairs that could be detected simultaneously with antibodies labelled with the same fluorophores, and their staining patterns later deconvolved using a feature extraction network previously trained. 3D volumetric imaging of a thick brain section is highlighted as validation of the method.

The idea of reducing the processing requirements of multiplexed imaging is indeed intriguing and valuable as there is a great need of original solutions aimed at mining datasets beyond traditional approaches. Yet, the solution proposed by the authors appears quite burdensome and of difficult implementation, especially for small plexes where other solutions are already available. The panel set up and training of the feature extracting network requires a significant commitment upfront (time and resources), which could only be justified for large cohorts of samples to be processed (i.e. large atlases) but not for more routinely used sample sizes.

We acknowledge that the initial implementation of our approach may appear complex. To address this concern, we delineate between initial setup and routine usage. The setup phase comprises two key steps: training of the feature extraction network to identify optimal grouping and training of the protein separation network. While this setup requires substantial initial investment, it is a one-time process that enables streamlined operations thereafter. In routine usage, our method delivers enhanced efficiency through simplified workflows - users benefit from pre-trained networks, single-round staining with optimized protein pairs, and reduced imaging time compared to traditional cyclic approaches. This efficiency compounds with each use, providing significant time and cost advantages over conventional methods.

We also note that, when a user intends to extend or change the protein palette, the singly-stained images from the previous experiments can be reused for training the feature extraction network. This indicates that the upfront cost is likely to remain manageable in many practical scenarios. We also verified that SEPARATE is compatible with the public human cell dataset from Allen Institute for Cell Science (Figure R1), which potentially allows us to use existing datasets.

Although we focused on demonstrating SEPARATE's proof-of-concept with a smaller protein panel, the approach demonstrates considerable scaling capabilities. Our implementation validates the method's robustness across various experimental conditions, with fundamental principles that extend readily to larger multiplexed panels. The modular architecture facilitates adaptation across different experimental scales, seamless integration with existing workflows, and enhanced capabilities for data reuse and meta-analysis.

Please note that, in cases where the comments from Reviewer #3 overlap with those from Reviewer #2, we have referenced the relevant comments from Reviewer #2 to avoid redundancy in this letter.

Additionally, the performance of the method is only shown for non-pathological samples in a tissue (brain) with very stereotypical expression patterns. It is not clear if SEPARATE would be applicable to the study of more complex sample cohorts where sourcing appropriate training data would be not trivial. This might be the case of patient cohort or embryonic development when expression patterns change as a result of disease or development for example , or in non-stereotypical samples such as tumour biopsies.

We acknowledge that the challenge of generalizing from stereotypical to non-stereotypical tissue patterns is indeed a fundamental limitation faced by all deep learning-based biomedical image analysis methods. At its core, this reflects the universal machine learning principle that networks require diverse training examples to handle varied scenarios effectively.

In this work, we focused on establishing a proof-of-concept for SEPARATE while demonstrating its robustness across different experimental conditions. Our cross-validation analysis (Figure R6) shows that SEPARATE maintains consistent performance across different dataset combinations, indicating robust feature extraction capabilities. Additionally, we validated SEPARATE using the public human cell dataset from Allen Institute for Cell Science containing diverse protein expression patterns (Figure R1). SEPARATE successfully handled various subcellular localizations and showed high correlation above 0.8 between feature-based distances and unmixing performance (We note that we are not looking for the correlation values that are close to one; these are for checking whether the feature distance is a good enough precursor of the unmixing performance). These results suggest that while SEPARATE was initially validated on brain tissue, its underlying principles are robust and applicable to diverse biological contexts.

Overall, we agree that applying SEPARATE to highly heterogeneous samples like tumor biopsies or developmental tissues would require careful consideration of training data. However, our validation results across different experimental conditions suggest that the method can adapt to various biological contexts when provided with appropriate training examples.

More should be added on the size and specs of the training dataset. From the information available in the Reporting Summary it appears to be extremely limited and arbitrary.

In line with the reviewer's comment, we have expanded the training dataset information in both the Methods section and the Reporting Summary, including detailed specifications on size, composition, and acquisition parameters.

The authors state that SEPARATE fills a void in the field by halving the processing time of cyclic imaging solutions, which they say range from 12hrs up to days long processing times. However, there are both academic protocols (e.g. HIFI, Watson S. et al, 2024 NatComm, <https://www.nature.com/articles/s41467-024-47185-9#Sec12>) and commercial solutions (Lunaphore COMET) with significantly shorter processing requirements (5-6 hrs/cycle for HIFI, 5hrs total for Lunaphore COMET).

Moreover, it is worth noting the existence of several commercial solutions able to achieve similar results. In example, Thermo Scientific has recently announced the release a new commercial spatial imaging solution (EVOS S1000) which used spectral deconvolution of optimised fluorophores to simultaneously detect up to 8 different antibody targets plus DAPI.

We appreciate the reviewer's thoughtful comments. As noted, there are multiple techniques and commercial instruments that facilitate multiplexed imaging. However, we would like to highlight two key points. First, prolonged staining times are required for specimens thicker than 10 μm . For example, recently published work on 3D multiplexed imaging of 35- μm -thick FFPE specimens reported staining durations of 8 to 10 hours (Yapp C, et al. Multiplexed 3D Analysis of Cell Plasticity and Immune Niches in Melanoma. *bioRxiv* (2023). doi:10.1101/2023.11.10.566670). For even thicker specimens, such as those 150 μm in thickness, staining may need to be extended to overnight or multiple days. Such prolonged staining times not only lengthen the overall process but also compromise specimen integrity and antigenicity. We believe that our technique, which reduces the number of cycles by half, provides a significant advantage in addressing these challenges. Accordingly, we have revised the introduction as follows:

“In two-dimensional imaging of thin tissue slices, such as those with a thickness of less than 10 μm , where staining can be completed relatively quickly—even in about an hour—the repeated staining process is not a major issue. However, specimens with a thickness beyond 10 μm , such as 35 μm , typically require much longer staining times, often extending up to 10 hours⁴. As a result, the total staining time for ten-color imaging could exceed several days. Prolonged staining time does not just extend the total process; it can also compromise specimen quality and antigen integrity^{5,6}. Spectral imaging and unmixing could provide a solution to this problem, as they allow the use of more than three fluorophores in a single staining cycle⁷. However,

spectral imaging can be complex and often requires specialized equipment capable of acquiring images from various spectral ranges^{8,9}. Therefore, there is an urgent need for techniques that can reduce the total staining time for 3D multiplexed imaging.”

As a general note, the method section is devoid of many key details (albeit some present in the Summary Report) that would impact the reproducibility of the method.

In line with the reviewer’s comment, we have expanded the Methods section to include detailed information about SEPARATE, incorporating content previously presented in the Summary Report. These additions provide the comprehensive details needed to facilitate the implementation and reproducibility of our method.

Similarly, it would be appreciated if the authors could release the code used for this manuscript.

We would like to clarify that our code had been publicly available on GitHub (<https://github.com/NICALab/SEPARATE>) since our initial submission, as stated in the "Code Availability" section of the manuscript.

More detailed comments below:

1. line 37. Imaging Mass Cytometry is not based on cyclical fluorescent imaging, yet is similarly adopted.

We have revised the text as suggested by the reviewer.

2. Line 107. The authors use synthetic images to generate training data for the feature extraction network. This is by linearly superimposing individual single staining images, circumventing the need to perform antibody optimization. However, this training dataset is methodologically different from the input data where the staining would have physically occurred simultaneously, and therefore does not take into account the behaviour of the antibodies when mixed together, for example competition, crowding, aspecificity or colocalization. It would be appreciated if the authors could comment on this or present data to address this issue.

We appreciate the reviewer’s thoughtful comments. As the reviewer pointed out, linearly superimposing individual single-staining images may differ from images acquired by staining specimens with multiple antibodies, primarily due to two factors: crowding and aspecificity.

First, crowding occurs when the binding of two antibodies to different targets is hindered due to the proximity of the targets. Literature suggests that such crowding effects are significant when target proteins are very close, which aligns with expectations. We agree that this effect could complicate the situation, and simply superimposing two images may not fully simulate this phenomenon. In our work, the proteins analyzed are predominantly cellular markers (e.g., PV, GFAP, NeuN, S100B, doublecortin) or proteins associated with specific organelles (e.g., lamin B1, MAP2, calnexin, nucleolin). These proteins are generally not expressed in close proximity, leading us to believe that the simulations in our study are unlikely to be affected by crowding effects. However, we acknowledge that caution is necessary when applying our technique to proteins that are closely localized, such as synaptic receptor proteins or proteins expressed in the same organelle. Second, regarding antibody crosstalk or aspecificity, we agree with the reviewer’s concerns. Most of the antibodies used in our study were validated for multiplexed imaging in our previous work (Seo et al., 2022, *Nat. Commun.* 13:2475). We found that combining these antibodies did not perceptibly alter staining patterns. Nevertheless, some antibodies may exhibit non-specific binding, and this must be carefully considered when designing antibody panels.

We have addressed these points in the discussion section with the following addition:

“The main limitation of our technique is that it assumes the expression patterns and levels acquired from singly stained specimens are identical to those of doubly stained specimens. This assumption holds only when antibodies bind to their targets in the same way when applied separately or together, meaning there is no significant crowding effect or antibody crosstalk. For crowding effects, studies report that antibody crowding is severe when target proteins are in close proximity¹⁰. The antibodies used in this study primarily target cellular markers or proteins expressed in different organelles, where crowding effects are likely minimal or negligible. However, when applying our technique to closely localized targets, such as synaptic receptor proteins within the same synapse or proteins expressed in the same organelle, careful studies on crowding are necessary. Regarding antibody crosstalk, the use of rigorously validated antibodies is essential. Antibodies available today undergo more stringent validation¹¹, but even so, careful evaluation of potential crosstalk is necessary when designing antibody panels for AI-based unmixing. This may involve additional testing to confirm that antibody combinations do not alter binding specificity or signal interpretation.”

3. Line 118. No mention is made in the text in reference to how many images are used to train the networks. There is some limited detail in the Reporting Summary, which should be added in the main text and more in detail in the methods. Additionally, the number of images stated is much more limited than other papers using feature extraction approaches on imaging datasets. Could the authors comment?

We thank the reviewer for raising these important points. We have revised the Methods section of the manuscript to provide more detailed information about the dataset used for training the network. For clarity, when we refer to a dataset, it represents a single z-stack image, consisting of multiple z slices. Each z-stack image has dimensions of 1024 (x) × 1024 (y) × 26 (z) pixels, corresponding to a volume of 204.8 μm × 204.8 μm × 25 μm. For each protein, we used three z-stack images for training the feature extraction network and one for identification of optimal grouping.

Regarding the dataset size comparison with other feature extraction approaches, while we acknowledge that a larger dataset could potentially enhance the performance of the network including its generalizability, the cross-validation analysis demonstrated that SEPARATE can effectively learn and extract meaningful features from this limited dataset, showing high reproducibility with a Pearson correlation coefficient of larger than 0.8 across different training and analysis dataset combinations. A detailed description of the cross-validation analysis and its results can be found in our response to Major Comment 9 of Reviewer #2 (Figure R7). We have incorporated these details in both the main text and the Methods section to ensure complete transparency and reproducibility of our approach.

4. The SSIM metric chosen by the authors quantifies across the entire field of view. It would be more appropriate if the authors could also provide a spatially aware metric of their methods to evaluate the accuracy across different cellular or tissue structures, position within the z stack and relative position within the field of view.

We thank the reviewer for this constructive suggestion regarding spatially aware evaluation metrics. We have addressed this concern by conducting additional spatial analysis across different regions and depths, as demonstrated in Figure R20 and R21.

This is now added as Supplementary Figures 8 and 9 in the supplementary materials.

Figure R20. Spatial analysis of unmixing results: group 1. a-e, For each protein pair in group 1, we selected representative z-positions from the complete z-stack and divided the field of view into a 4×4 grid to evaluate local unmixing accuracy. From left to right: grayscale input image, unmixed result, ground truth image, and grid-wise SSIM map. Both unmixed and ground

truth images are shown as two-channel merges using a green-magenta color scheme. **a**, Results of (calbindin 2, calnexin) pair, with calbindin 2 in green and calnexin in magenta; **b**, Results of (doublecortin, nucleolin) pair, with doublecortin in green and nucleolin in magenta; **c**, The results of (GFAP, NeuN) pair, with GFAP in green and NeuN in magenta; **d**, The results of (lamin B1, PV) pair, with lamin B1 in green and PV in magenta; **e**, The results of (MAP2, S100B) pair, with MAP2 in green and S100B in magenta. Scale bar = 30 μm in **a-e**.

Figure R21. Spatial analysis of unmixing results: group 2. a-e, For each protein pair in group 2, we selected representative z-positions from the complete z-stack and divided the field of view into a 4×4 grid to evaluate local unmixing accuracy. From left to right: grayscale input image, unmixed result, ground truth image, and grid-wise SSIM map. Both unmixed and ground

truth images are shown as two-channel merges using a green-magenta color scheme. **a**, Results of (calbindin 2, PV) pair, with calbindin 2 in green and PV in magenta; **b**, Results of (calnexin, lamin B1) pair, with calnexin in green and lamin B1 in magenta; **c**, Results of (doublecortin, MAP2) pair, with doublecortin in green and MAP2 in magenta; **d**, Results of (GFAP, S100B) pair, with GFAP in green and S100B in magenta; **e**, Results of (NeuN, nucleolin) pair, with NeuN in green and nucleolin in magenta. Scale bar = 30 μm in **a-e**.

5. Line 150. The authors should include in the text the thickness of the sample and details of the imaging. In the current wording “volumetric” could be easily interpreted as applied to larger samples than a tissue section.

We have modified the manuscript to specify the imaging parameters: "While our tissue sections are 150 μm thick, imaging was performed to a depth of approximately 25 μm due to antibody penetration limitations and signal-to-noise considerations in deeper tissue regions." These specifications have been added to both the Methods section and main text to ensure clarity regarding the scale of our three-dimensional analysis.

6. Line 168. The authors should validate their method on more than a single sample and provide a metric to demonstrate reproducibility, ideally across different tissue types and different staining.

To address the reviewer's concern about reproducibility across multiple samples, we refer to our response to major comment 9 of reviewer #2, where we present a comprehensive cross-validation analysis using four independent datasets. As demonstrated in Figure R3, our method shows high reproducibility with correlation coefficients between the feature distance and SSIM consistently above 0.8 across different dataset combinations. We note that we are not looking for the correlation values close to one, as these numbers are for simply checking whether the feature distance is a good enough precursor of the unmixing performance.

7. Line 180. The authors should increase the sample size and diversity of tissues/antigens in order to comment on the robustness.

In line with the reviewer's comment, we have assessed the robustness of our method using the public human cell dataset from Allen Institute for Cell Science containing 10 proteins with different subcellular localizations. We focused on our key idea of quantifying the relationship between feature-based distances and unmixing performance for identifying optimal protein grouping. Our analysis validates the feature-based distance as a reliable metric for predicting unmixing success, showing strong correlations between feature-based distances and unmixing performance—Pearson correlations of 0.8225 and 0.8853—for protein pairs with cell membrane and DNA, respectively. For a comprehensive analysis of our validation across multiple protein combinations, including detailed unmixing performance metrics and visualization results, we refer the reviewer to our response to major comment 3 of reviewer #2 (Figures R1-3), where we demonstrate the robustness of our method across different proteins and staining conditions.

8. Line 197. This claim is not fully supported by the data presented as the thickness of the sample shown in the manuscript is significantly smaller (150 μm), and this could impact the performances of the stainings and therefore of the algorithms.

We appreciate the reviewer's concern regarding tissue thickness. While our tissue sections are 150 μm thick as described in the Methods section, we utilized approximately 25 μm depth for imaging due to the limited signal-to-noise ratio in deeper regions caused by antibody penetration constraints. This is standard practice in immunofluorescence imaging, where effective imaging depths are typically less than the total slice thickness.

Our results show that unmixing performance improves with increasing z-stack depth (Figure R12), suggesting that deeper tissue imaging through optimized staining protocols would yield even better results. Since staining occurs bidirectionally from both surfaces of the slice, our method would be readily applicable to thinner sections, such as 50 μm slices, where complete staining penetration requires only 25 μm from each surface.

9. Moreover, it would be good if the authors could comment on the portability of SEPARATE to samples imaged with other methods, such as widefield or light sheet microscopy, which are commonly used in other multiplexed imaging settings.

We appreciate this insightful comment about the compatibility of SEPARATE with other imaging modalities. While we have not experimentally validated SEPARATE with widefield or light sheet microscopy, we expect our method to be fundamentally compatible with these systems. However, it is important to note that the performance might be affected by the presence of out-of-focus light, which is inherent in both widefield and light sheet microscopy. This background signal could potentially reduce the accuracy of our method compared to its performance with confocal microscopy. We have incorporated this discussion about the potential usage of other imaging modalities in the Discussion section of the manuscript.

10. Line 198. The authors could include the time required to set up SEPARATE from choosing the optimised antibody pairs to training the feature extraction algorithm and running it on imaging datasets.

Thank you for your suggestions. We will clarify the time requirements by separating them into two main categories: one-time setup process and routine application.

The one-time setup process comprises several essential steps. First, we perform immunofluorescence staining of proteins for pairing optimization, requiring standard staining time. Next, we train the feature extraction network and identify optimal grouping, which takes approximately one hour. For the identified optimal pairs, we prepare specialized training data for the protein separation network. This requires another round of immunofluorescence staining to create two separate channels, each containing one protein from the identified pair. Finally, we train the protein separation network, which takes approximately 10 hours. While this setup process requires significant time investment, it only needs to be performed once.

For routine applications, our method becomes remarkably efficient by reducing the imaging time compared to conventional cyclic imaging methods. Users simply need to stain their samples with identified optimal protein pairs for each single fluorophore, capture images, and process them through our pre-trained networks. This streamlined approach eliminates the need for multiple rounds of staining and imaging cycles, making our method increasingly cost-effective with each subsequent use. With repeated use, the time savings from our method will substantially outweigh the initial setup time investment.

We now have incorporated this to the Discussion section of the manuscript.

11. Line 229 (figure 2c). The scale bar is missing values.

We have updated Figure 2c with the actual values for the colormap of the feature-based distance matrix.

12. Line 234 (Figure 3c, 3d). This would be better substituted with supplementary figure 2. Although the consistency with the tSNE coloring scheme is appreciated, it negatively impacts the readability of the figure, as some of the color pairs cannot be easily discerned (for example in the calbindin 2/calnexin or NeuN/nucleolin). Ideally this should be color-blind friendly.

We have updated the coloring scheme for Figure 3c,d to improve readability and ensure color-blind friendliness, substituting it with Supplementary Figure 2, which uses a green-and-magenta color scheme.

13. Line 246 (Figure 4c-e). could the authors label images with the proteins displayed similarly to 4a?

We have added labels to indicate the proteins displayed in the respective images for Figure 4c-e in the revised version.

14. Line 260 and 278. No details of the antibody used and succinimidyl ester-fluorophores used. Similarly, the authors should state the actually primary antibody concentrations used

We have included details of the reagents and the antibodies used in the Supplementary Tables 1 and 2, and the primary antibody concentration in the Method section.

15. Line 291. The authors should provide more details on the imaging set up, for example pixel size, presence of any averaging step, spacing of the optical sections in the z-stacks.

We have revised the Methods section to provide the details on the fluorescent imaging setup. Specifically, we have included the pixel size of $0.2 \mu\text{m} \times 0.2 \mu\text{m}$, stated that no averaging steps were performed during image acquisition, and specified a spacing of either 0.5 or 1 μm between optical sections in the z-stacks.

REFERENCES

- 1 Ounkomol, C., Seshamani, S., Maleckar, M. M., Collman, F. & Johnson, G. R. Label-free prediction of three-dimensional fluorescence images from transmitted-light microscopy. *Nature methods* **15**, 917-920 (2018).
- 2 Woolfe, F., Gerdes, M., Bello, M., Tao, X. & Can, A. Autofluorescence removal by non-negative matrix factorization. *IEEE Transactions on image processing* **20**, 1085-1093 (2010).
- 3 Brown, J. K., Pemberton, A. D., Wright, S. H. & Miller, H. R. Primary antibody–Fab fragment complexes: a flexible alternative to traditional direct and indirect immunolabeling techniques. *Journal of Histochemistry & Cytochemistry* **52**, 1219-1230 (2004).
- 4 Yapp, C. *et al.* Multiplexed 3D Analysis of Cell Plasticity and Immune Niches in Melanoma. *bioRxiv*, 2023.2011.2010.566670 (2023).
- 5 Mascadri, F., Bolognesi, M. M., Pilla, D. & Cattoretti, G. Rejuvenated vintage tissue sections highlight individual antigen fate during processing and long-term storage. *Journal of Histochemistry & Cytochemistry* **69**, 659-667 (2021).
- 6 Karlsson, C. & Karlsson, M. G. Effects of long-term storage on the detection of proteins, DNA, and mRNA in tissue microarray slides. *Journal of Histochemistry & Cytochemistry* **59**, 1113-1121 (2011).
- 7 Zimmermann, T., Marrison, J., Hogg, K. & O’Toole, P. Clearing up the signal: spectral imaging and linear unmixing in fluorescence microscopy. *Confocal microscopy: methods and protocols*, 129-148 (2014).
- 8 Cohen, S., Valm, A. M. & Lippincott-Schwartz, J. Multispectral live-cell imaging. *Current protocols in cell biology* **79**, e46 (2018).
- 9 Seo, J. *et al.* PICASSO allows ultra-multiplexed fluorescence imaging of spatially overlapping proteins without reference spectra measurements. *Nature Communications* **13**, 2475 (2022).
- 10 De Michele, C., De Los Rios, P., Foffi, G. & Piazza, F. Simulation and theory of antibody binding to crowded antigen-covered surfaces. *PLoS computational biology* **12**, e1004752 (2016).
- 11 Thul, P. J. & Lindskog, C. The human protein atlas: a spatial map of the human proteome. *Protein Science* **27**, 233-244 (2018).

Response to review

We thank the reviewers for their assessment of our revised submission. The manuscript has improved thanks to the reviewers' thorough feedback. We have addressed all points raised during the review process. Our detailed responses to each reviewer's comments are provided below, with corresponding changes implemented throughout the manuscript.

Below is a summary of the major changes in response to the review comments:

1. We have extended our evaluation to show unmixing performance:
 - Incorporated both SSIM and PCC as performance metrics across all datasets, presenting these results in the main figures instead of supplementary materials.
 - Conducted statistical test comparing optimal versus alternative grouping (Supplementary Figure 17).
 - Included unmixing performance for individual proteins of paired protein (Supplementary Figures 13 and 24).
2. We have expanded the methodological details and validation and modified visualization methods for more accurate representation
 - Added detailed formulas for calculating unmixing performance.
 - Replaced channel-wise merged images with single channel grayscale images.
3. We have expanded the discussion of practical implementation considerations:
 - Added discussion on the limitations of finding optimal grouping without co-localization information and practical considerations for obtaining robust feature vectors.
 - Included quantitative thresholds for acceptable feature-based distances.
 - Addressed the limited signal-to-noise ratio caused by antibody penetration constraints and the use of bidirectional staining to obtain deeper imaging depth.

Reviewer #1 (Remarks to the Author):

The authors have addressed the comments and concerns that the reviewers made in a detailed manner. They have now also included additional figures in the main and the supplemental that explain more details of the methodology. The limitations have been also added to the main manuscript. There are no further comments from my side.

We thank the reviewer for their positive assessment and confirmation that our revisions have addressed their concerns. The reviewer's thoughtful guidance has strengthened the structure and clarity of our manuscript.

Reviewer #2 (Remarks to the Author):

The authors have made a significant effort in addressing many of the points raised during the initial review. The inclusion of the Allen Cell dataset, additional figures, and clarifications in the methods section have strengthened the manuscript. The analysis of different factors, such as the impact of resolution and noise of the reference images are also important new insights.

However, several concerns remain that need further clarification, along with some technical and presentation-related improvements to enhance the robustness and clarity of the study.

We thank the reviewer for acknowledging our efforts to address the points raised during the initial review and for the positive feedback on our manuscript enhancements. We have carefully addressed the remaining concerns in our detailed responses below.

Regarding unmixing performance, we have incorporated both SSIM and PCC as performance metrics across all datasets and elevated this information to the main figures rather than supplementary materials. We also presented unmixing performance for individual channels (please see major comment 2), performed statistical tests comparing optimal versus alternative groupings to strengthen our analysis (please see major comment 3), and added detailed formulas for calculating unmixing performance to provide clarity. We have revised our data visualizations throughout the manuscript in accordance with the reviewer's suggestions.

Major Comments

1. Interpretation of Allen Cell Dataset Results: The results from the Allen Cell dataset raise concerns regarding the practical usability of the method. Intuitively, non-spatially overlapping markers like DAPI and ZO1 should have the highest separability; however, these markers performed poorly in the feature distance matrix. This discrepancy warrants further investigation and explanation.

We appreciate the reviewer's insightful observation regarding the Allen Cell dataset results. This comment touches on an important aspect of our methodology that warrants clarification.

Our feature vector extraction process does not explicitly incorporate information about the actual spatial overlap between proteins. Although incorporating such spatial relationship data could theoretically enhance performance, it would require prior imaging data for all possible protein pair combinations, which would be prohibitively expensive and would defeat the purpose of our approach. SEPARATE is specifically designed to determine optimal protein pairings using only individual protein images, without requiring prior knowledge of how protein pairs co-localize.

In examining the specific case of ZO-1 and DNA markers in the Allen Cell dataset, we identified several dataset-specific factors that influenced feature vector extraction. The ZO-1 dataset shows clear tight junction structures in only a small fraction of z-planes, with most planes displaying unclear signals. Similarly, the DNA marker images contain relatively few well-focused planes and several out-of-focus, blurry regions. When examining the corresponding images of markers positioned close together in our feature space, we observed that both exhibit significant noise and unclear patterns (Figure R1). These observations highlight why intuitively non-spatially overlapping markers like DAPI and ZO1 may still perform poorly in our feature distance matrix.

As SEPARATE enables efficient selection of optimal protein pairs without individually imaging numerous protein combinations, common background characteristics, such as blurry or diffuse signals, can themselves function as important features rather than merely being considered noise. These common patterns can be interpreted as signals indicating "potential

difficulty in unmixing," and SEPARATE aims to preemptively identify which pairs should avoid being mixed together versus which pairs can be effectively unmixed. Therefore, in situations where direct spatial co-localization information is unknown, feature-based classification that considers background similarities remains strategically valid for predicting unmixing challenges. Although not currently considered in our framework, incorporating prior knowledge about subcellular localization such as which organelle a protein is expressed in could potentially yield more reliable feature vectors even without direct imaging data.

Figure R1. Visualization of protein images that are closely located in the feature domain. a, The t-SNE plot of extracted feature vectors from each protein image patch and boxed regions showing protein pair that are closely located in the feature domain. **b,** Image patches corresponding to the boxed region in **a.** ZO-1 and DNA. Scale bar = 5 μm in **b.**

2. Reporting SSIM: In Figure 3b, 5d-e and Supplementary Fig. 19, it is unclear which channel was used for calculating the SSIM values. How is the SSIM calculated for the unmixed versus groundtruth? I'm guessing single-channel images were used for this, and in this case SSIM should have been reported for both of the channels. It is not specified which one we are seeing in panels d and e, but I am guessing it is the SSIM for the markers in the x-axis, and for the cell membrane or DAPI. However, the similarity might be quite different if they quantify the SSIM and PCC for cell membrane or unmixing for each pair, which would actually make it possible to understand how the pairing is affecting the unmixing of the same protein in each case.

If my assumption is wrong and the SSIM is being reported for multi-color images, I do not think this is appropriate. Additionally, the manuscript utilizes SSIM almost exclusively for evaluating the unmixing performance. SSIM (between unmixed and groundtruth) might not be the best stand alone metric for evaluating the unmixing performance, especially if the aim is seeing how well a marker is separated from the other (which was also partially raised in Point 12 before). In their calculation SSIM values appear quite high for most conditions (above 0.7 in most cases), which suggests it is not very sensitive to critical changes in spatial distribution. Other correlation coefficients like PCC or Mander's are expected to be more suitable in this case to show reliable separation (rather than similarity which also looks at different factors that are not as relevant here), and all of the unmixing performance measurements need to be done separately for both channels that are being unmixed and reported as such. They have now included Supplementary Fig. 19 with PCC calculations, but as noted above it is unclear how these are calculated for pairs and what is being reported. Similarly, not much explanation is given about how cross-SSIM or cross-PCC is calculated. This is also only done for one of the main experiments, and is included only as a supplement.

We appreciate the reviewer's thorough of unmixing performance. To clarify the concerns raised:

Regarding performance metrics, we have implemented both SSIM and PCC as performance metrics for all datasets, now presenting these results in the revised main Figures 3 and 5, rather than relegating this information to supplementary materials. For each protein pair, the unmixing performance metrics (both SSIM and PCC) were previously calculated separately for each individual protein, and the final value was obtained by averaging these two values. We would like to clarify that we did not report unmixing performance for only one protein channel, but rather used the average value of both channels. However, we agree with the reviewer that unmixing performance should be presented for each channel separately, and we have included this detailed information in Figure R2 and R3, showing the unmixing performance of individual channels for both SSIM and PCC. Additionally, we acknowledge that our method of calculating the average unmixing performance across channels was not clearly specified, and we have added this detail to the Method section.

Furthermore, as the reviewer correctly pointed out, our explanation was insufficient regarding the calculation for cross-SSIM and cross-PCC, which are used to evaluate cross-similarity between two target proteins being unmixed. We have now added more detailed explanations and formulas in the Method section.

This is now added as Supplementary Figures 13 and 24 in the supplementary materials.

Figure R2. Unmixing performance of individual proteins. a, Comparison of between feature-based distance and unmixing performance using SSIM and PCC as metrics. Protein pairs belonging to optimal grouping, group 1, are labeled in black on the axis, while protein pairs belonging to alternative grouping, group 2, are labeled in red. For each protein pair, unmixing performance of individual proteins were displayed as box-and-whisker plots for non-overlapping patches ($n = 16$), with the feature-based distances shown as a purple x marker. The boxes show the interquartile range (IQR) with the median, while whiskers extend to 1.5 times the IQR. Individual points represent outliers, defined as values falling outside the whiskers.

Figure R3. Unmixing performance of individual proteins for the public human cell dataset from Allen Institute for Cell Science. a-b, Comparison of between feature-based distance and unmixing performance using SSIM and PCC as metrics. For each protein pair, unmixing performance of individual proteins were displayed as box-and-whisker plots for non-overlapping patches ($n = 16$), with the feature-based distances shown as a purple x marker. The boxes show the interquartile range (IQR) with the median, while whiskers extend to 1.5 times the IQR. Individual points represent outliers, defined as values falling outside the whiskers. **a,** Unmixing performance of 8 proteins paired with membrane, where the left box in each pair shows the unmixing performance of the protein paired with membrane and the right box shows the unmixing performance of the membrane; **b,** Unmixing performance of 8 proteins paired with DNA, where the left box in each pair shows the unmixing performance of the protein paired with DNA and the right box shows the unmixing performance of the DNA.

3. Additionally, the manuscript lacks statistical significance testing for the SSIM values of different protein pairs, which generally appear quite high for most conditions (for example in Fig. R1d-e or Fig. R10). Reporting statistical comparisons would improve the interpretability of these results and help understand the actual improvement between pairs that have a high distance and low distance in the feature-based distance matrices.

There should also always be a negative control case, where similarity/correlation is expected to be naturally low. For calculating statistics authors refer to 16 patches obtained from the same image as “independent” patches - these are not independent.

Following the reviewer’s suggestion, we have performed statistical tests comparing unmixing performance between two grouping. The optimal grouping, group 1 consists of the following five pairs: (calbindin 2, calnexin), (doublecortin, nucleolin), (GFAP, NeuN), (lamin B1, PV), and (MAP2, S100B). The alternative grouping, group 2 consists of the following pairs: (calbindin 2, PV), (calnexin, lamin B1), (doublecortin, MAP2), (GFAP, S100B), and (NeuN, nucleolin).

To assess statistical significance, we compared the unmixing performance (both SSIM and PCC metrics) of each protein when it appeared in optimal grouping versus alternative grouping (Figure R4a). We conducted a two-sided paired-sample t-test between unmixing performance values of the same protein in different groupings. While not all proteins showed statistically significant differences, we found that in all cases where significant differences were observed, the protein in optimal grouping exhibited higher unmixing performance compared to the alternative grouping. For SSIM, significant differences were found in lamin B1 ($p < 0.1$), doublecortin, GFAP, nucleolin ($p < 0.01$), and MAP2 ($p < 0.001$). For PCC, significant differences were found in GFAP ($p < 0.01$). We note that, by design, our grouping strategy does not seek to improve unmixing performance across all proteins, as explained below in the manuscript:

“Our criterion for identifying optimal grouping by maximizing the minimum distance among paired proteins ensures robust unmixing, even for the most challenging protein combinations. This approach emphasizes consistent performance across all protein pairs rather than allowing exceptional unmixing in some pairs at the expense of others.”

Additionally, we performed statistical test for normalized unmixing performance, which adjusts the raw performance by the cross-similarity between paired proteins (Figure R4b). Normalized unmixing performance showed a more significant difference between optimal grouping and alternative grouping. For SSIM, significant differences were observed in GFAP, MAP2 ($p < 0.1$), NeuN and nucleolin ($p < 0.001$). For PCC, significant differences were found in PV, S100B ($p < 0.1$), calnexin ($p < 0.01$), doublecortin, GFAP, lamin B1, NeuN, and nucleolin ($p < 0.001$).

Regarding the reviewer’s concern about patch independence, we acknowledge that non-overlapping patches obtained from the same image are not perfectly independent. However, non-overlapping patches can be interpreted as a different field of view within the same sample, and we carefully selected separate regions to optimize independence within our dataset. For the experiment using the Allen Cell Dataset, we maintained independence by using different individual images.

This is now added as Supplementary Figures 17 in the supplementary materials.

Figure R4. Statistical comparison of unmixing performance between optimal grouping (group 1) and alternative grouping (group 2). **a**, Comparison of unmixing performance using SSIM (top) and PCC (bottom) for each protein in different groupings. Box-and-whisker plots show the unmixing performance values across non-overlapping patches ($n = 16$). A two-sided paired-sample t-test was used. **b**, Comparison of unmixing performance using normalized SSIM (top) and normalized PCC (bottom), which adjusts raw performance by the cross-similarity between paired proteins, for each protein in different grouping. Box-and-whisker plots show the unmixing performance values across non-overlapping patches ($n = 16$). A two-sided paired-sample t-test was used. ($*p < 0.1$, $**p < 0.01$, $***p < 0.001$).

4. Boxplot Visualization (for eg. Figures 3b and 5d-e and Supplementary Fig. 19): The feature-based distance is presented as a continuous dotted line, which makes it difficult to see the actual values. Since these distances represent distinct spots, connecting them with lines can be misleading. I recommend using distinct markers without connecting the data points. Also, I would recommend plotting the boxes in the order of decreasing/increasing distances (or SSIM), so one could see the trend more clearly.

Median vs. Average in Line 330: The authors mention using the median in the boxplots but calculate PCC using the average SSIM. This discrepancy should be resolved, and the choice of metric should be clearly justified.

It seems in both figures there are some outliers (which are quite hard to see with the current label). Especially for Fig. 3, which patches are regarded as outliers (since they are all shown in Supplementary Fig. 8 and 9), and how are they justified.

For the boxplot visualization, we have modified Figures 3b, 5d-e, and Supplementary Figure 16 (previously, Supplementary Figure 19) to display data points as individual markers without connecting line. Additionally, we have rearranged the boxes in order of decreasing feature-based distance values to make trends more readily apparent.

Regarding the discrepancy between median and average values, boxplots were used to show the five-number summary (sample minimum, lower quartile, median, upper quartile, sample maximum) as is standard practice. Following the reviewer's suggestion, we have now included PCC calculations using both mean and median SSIM values to ensure comprehensive and consistent statistical reporting.

For the outliers in box-and-whisker plots, we have enhanced their visibility in our revised visualizations. Furthermore, we have added explanations in the figure captions stating that outliers are defined as values falling outside the whiskers, which extend to 1.5 times the interquartile range (IQR).

5. The two-color overlays used in many figures (for eg. Supplementary Fig. 3-8) are not helpful to have a reliable visual comparison of unmixed images to ground truth images, these should be presented as single by side single channel grayscale images.

Following the reviewer's recommendation, we have now modified figures to display unmixed images and ground truth images as side-by-side single channel grayscale images rather than as overlays.

6. Figure R5 clearly shows that the cell density in the image patches is a crucial factor for feature vectors, but the authors do not discuss the implications of this observation. This is an important discussion to understand the limitations of the approach.

As such, insufficient sampling of the patches (for example having lots of empty tissue areas included in the input dataset) could yield very different feature vectors.

We appreciate the reviewer's comment regarding the relationship between cell density and feature vectors, as demonstrated in Figure R5 (Supplementary Figure 2). We acknowledge that this important consideration was not sufficiently discussed in the original manuscript.

Our finding shows that for nucleolin and NeuN, image patches from overlapping cluster regions show lower cell density, limiting the information available to distinguish these proteins. This challenge occurs because these proteins share similar expression patterns, making differentiation difficult in low-density areas where distinctive features are not sufficiently captured. In contrast, proteins with distinct expression patterns, such as GFAP and NeuN, can be expected to remain distinguishable even in low-density regions as they lack common expression patterns. In other words, while cell density may become a limiting

factor for distinguishing proteins that share similar expression patterns, it likely has minimal impact on the differentiation of proteins with distinct expression patterns.

Additionally, we agree with the reviewer's point that insufficient sampling, particularly including numerous empty tissue areas in the input dataset, can lead to substantially different feature vectors. This is a valid concern for any feature extraction methodology applied to microscopy data. However, the presence of empty space in images is not inherently problematic for our method. SEPARATE is designed to achieve reliable unmixing not only in regions with clearly defined expression patterns but across the entire tissue, including areas considered as background. In this context, faint expressions in background regions are treated as low-level features and incorporated into the learned representation. This provides a more comprehensive and robust basis for predicting which protein pairs are likely to be effectively unmixed under diverse tissue conditions. Importantly, in the absence of prior information such as co-localization or subcellular localization, background similarity can serve as a valid proxy for estimating unmixing performance. While background similarity does not universally determine unmixing success or failure, it can serve as a reasonable proxy in the absence of stronger differentiating cues.

As the reviewer correctly points out, this approach requires sufficient sampling to be effective. In our context, sufficient sampling refers to how well the model is exposed to varying ratios of empty and signal-rich regions during training. To address this, we randomly crop multiple patches from images that are larger than the network input size, producing a diverse dataset with different combinations of cellular structures and background space. This strategy introduces subtle variations even across adjacent regions and allows the model to learn robust features under heterogeneous tissue conditions. As a result, SEPARATE can better mitigate the potential impact of insufficient sampling and generalize to variable experimental conditions.

We have added this discussion to the manuscript.

Minor Comments:

- Supplementary Figure 8 - how local unmixing accuracy was calculated is not explained in the legend.

We thank the reviewer for pointing out this omission. We apologize for the confusion in our terminology, as we inconsistently referred to 'local unmixing accuracy' when we meant 'local unmixing performance' in our figure descriptions. In the revised caption for Supplementary Figures 14 and 15 (previously Supplementary Figures 8 and 9), we have clarified how local unmixing performance was calculated and corrected this terminology. Specifically, we divided each image into a 4×4 grid (16 cells) and calculated the Structural Similarity Index Measure (SSIM) between the unmixed result and ground truth image for each grid cell. The SSIM values, ranging from -1 to 1, are represented in the grid-wise SSIM maps shown in the figure.

- Thickness: In the methods section authors state “patches of size 512(x) × 512(y) × 16(z) were extracted from the input image stack”. 16 z sections mean either 8 or 16 μm thick z-stack, which is smaller than what they state in the methods. Could the authors provide additional reasoning which z planes were removed and why the z-stack was reduced from 25μm to 8 or 16 μm?

We would like to clarify that our original full-size image stack has dimensions of 1024(x) × 1024(y) × 25(z), and we utilized randomly cropped patches of size 512(x) × 512(y) × 16(z) from this full image stack for training the protein separation network. The specific choice of 16 for the z-dimension was determined by the architectural requirements of the protein separation network, which utilizes max pooling operations along the z-axis. To ensure compatibility with these operations, we needed to use a power of 2 for the z-dimension. Rather than selectively reducing or removing specific z-planes from the whole image, we instead extracted appropriately sized random patches from the full volume.

- Introductory Statement (Line 44): The authors should clarify their statement regarding staining duration, as methods like DNA oligo barcoding allow for much shorter cycles.

We thank the reviewer for pointing this out. We have changed the sentence to clarify that such time requirement is for conventional immunostaining approaches that use full IgG.

- Synthetic Image Representation (Figure R2): Synthetic images should clearly label both markers, as they are mixtures of the respective markers. The manuscript does not specify which color corresponds to which marker. The coefficients used for generating synthetic images should be added to the methods section.

While we had included the information about which color corresponds to which protein marker in the caption of Figures R2 and R3 (Supplementary Figures 22 and 23), we acknowledge that having this information directly on the figure would improve clarity. We have revised Supplementary Figures 22 and 23 to include clear labels directly on the images that explicitly indicate which color corresponds to which marker.

Regarding the coefficients used for generating synthetic images, we used 0.5 for both α and β and added this information to the manuscript.

- Dataset Clarification (Figure R4): The figure does not clearly indicate which dataset was used for training and which for testing. Furthermore, it should be stated whether any FOVs were reused in both datasets.

We have added explicit information to the caption of Figure R4 (Supplementary Figure 18) to clearly indicate which datasets were used for training and testing. Specifically, for each protein, we used three z-stacks for training data and one z-stack for test data, ensuring complete separation between them.

- Grayscale Presentation: Single-channel microscopy images should consistently be presented in grayscale for better interpretability.

We would like to clarify that in our previous revision, we had already updated all experimental single-channel microscopy images in grayscale. However, we noticed that the schematic figure (Figure 1b) containing single-channel representations were not consistently following this convention. In response to the reviewer's comment, we have now updated the schematic figure as well to ensure that all single-channel data, whether in experimental images or schematics, are consistently presented in grayscale throughout the entire manuscript and supplementary materials.

Additionally, as stated in major comment 5, we have now modified figures to display unmixed images and ground truth images as side-by-side single channel grayscale images rather than as overlays.

- Figure 3 Overlays: The main results in Figure 3 should be zoomed in to improve visibility.

We have updated the Figure 3 to improve visibility of the result images.

- Feature-Based Distance Thresholds: Quantitative thresholds for acceptable feature-based distances should be provided to assist researchers in applying SEPARATE to their own datasets.

Since our feature vectors are latent variables, their absolute values are less meaningful than their relative relationships. To address this, we normalized the feature space so that all feature vectors lie within a unit circle (as in Supplementary Figure 18). Through our extensive experiments, we observed that when the normalized feature-based distance is greater than 1.5, achieves unmixing performance with SSIM values above 0.9. We have incorporated this to the Discussion section of the manuscript.

- Figure 4 Labeling: Protein names should be explicitly labeled in all sub-panels.

We have added protein names to the respective images for Figure 4f in the revised version.

- Use of Fiducial Markers: The authors suggest using DAPI as a fiducial marker but mention it being in the 488 channel instead of 405 nm. This needs correction.

We thank the reviewer for identifying this error. The reviewer is correct that DAPI is visualized in the 405 nm channel, not the 488 nm channel as we incorrectly stated. We have corrected this technical inaccuracy throughout the manuscript to properly indicate that DAPI is used as a fiducial marker in the 405 nm channel.

- Heatmap Values: Numeric values would ideally be added to the heatmap for better interpretability.

We have updated Figures 2 and 5, as well as Supplementary Figure 13 to show numeric values directly on the heatmaps.

- Formula Documentation: Formulas for metrics like cross-SSIM should be clearly documented.

In response to this comment, we have added a description of cross-similarity (cross SSIM and cross PCC) and normalized performance (normalized SSIM and normalized PCC), including its mathematical formula, to the Methods section of our revised manuscript.

Response to review

We are grateful for the reviewers' careful review of our revision. Their detailed feedback has contributed to improving the manuscript.

Reviewer #2 (Remarks to the Author):

The authors have addressed all my comments in the last revision, which improved the transparency and accurate presentation of the data and the quality of the discussion. I do not have further concerns.

We thank the reviewer for thorough review and constructive feedback throughout the revision process. We appreciate reviewer's time and effort in helping us improve the manuscript.